# $b$-bit Marginal Regression

**Martin Slawski**
Department of Statistics and Biostatistics
Department of Computer Science
Rutgers University
martin.slawski@rutgers.edu

**Ping Li**
Department of Statistics and Biostatistics
Department of Computer Science
Rutgers University
pingli@stat.rutgers.edu

## Abstract

We consider the problem of sparse signal recovery from $m$ linear measurements quantized to $b$ bits. $b$-bit Marginal Regression is proposed as recovery algorithm. We study the question of choosing $b$ in the setting of a given budget of bits $B = m \cdot b$ and derive a single easy-to-compute expression characterizing the trade-off between $m$ and $b$. The choice $b = 1$ turns out to be optimal for estimating the unit vector corresponding to the signal for any level of additive Gaussian noise before quantization as well as for adversarial noise. For $b \geq 2$, we show that Lloyd-Max quantization constitutes an optimal quantization scheme and that the norm of the signal can be estimated consistently by maximum likelihood by extending [15].

## 1 Introduction

Consider the common compressed sensing (CS) model

$$y_i = \langle a_i, x^* \rangle + \sigma \varepsilon_i, \quad i = 1, \ldots, m, \quad \text{or equivalently}$$
$$y = Ax^* + \sigma \varepsilon, \quad y = (y_i)_{i=1}^m, \quad A = (A_{ij})_{i,j=1}^{m,n}, \quad \{a_i = (A_{ij})_{j=1}^n\}_{i=1}^m, \quad \varepsilon = (\varepsilon_i)_{i=1}^m, \tag{1}$$

where the $\{A_{ij}\}$ and the $\{\varepsilon_i\}$ are i.i.d. $N(0,1)$ (i.e. standard Gaussian) random variables, the latter of which will be referred to by the term "additive noise" and accordingly $\sigma > 0$ as "noise level", and $x^* \in \mathbb{R}^n$ is the signal of interest to be recovered given $(A, y)$. Let $s = \|x^*\|_0 := |S(x^*)|$, where $S(x^*) = \{j : |x_j^*| > 0\}$, be the $\ell_0$-norm of $x^*$ (i.e. the cardinality of its support $S(x^*)$). One of the celebrated results in CS is that accurate recovery of $x^*$ is possible as long as $m \gtrsim s \log n$, and can be carried out by several computationally tractable algorithms e.g. [3, 5, 21, 26, 29].

Subsequently, the concept of signal recovery from an incomplete set ($m < n$) of linear measurements was developed further to settings in which only coarsely quantized versions of such linear measurements are available, with the extreme case of single-bit measurements [2, 8, 11, 22, 23, 28, 16]. More generally, one can think of $b$-bit measurements, $b \in \{1, 2, \ldots\}$. Assuming that one is free to choose $b$ given a fixed budget of bits $B = m \cdot b$ gives rise to a trade-off between $m$ and $b$. An optimal balance of these two quantities minimizes the error in recovering the signal. Such optimal trade-off depends on the quantization scheme, the noise level, and the recovery algorithm. This trade-off has been considered in previous CS literature [13]. However, the analysis therein concerns an oracle-assisted recovery algorithm equipped with knowledge of $S(x^*)$ which is not fully realistic.

In [9] a specific variant of Iterative Hard Thresholding [1] for $b$-bit measurements is considered. It is shown via numerical experiments that choosing $b \geq 2$ can in fact achieve improvements over $b = 1$ at the level of the total number of bits required for approximate signal recovery. On the other hand, there is no analysis supporting this observation. Moreover, the experiments in [9] only concern a noiseless setting. Another approach is to treat quantization as additive error and to perform signal recovery by means of variations of recovery algorithms for infinite-precision CS [10, 14, 18]. In this line of research, $b$ is assumed to be fixed and a discussion of the aforementioned trade-off is missing.

In the present paper we provide an analysis of compressed sensing from $b$-bit measurements using a specific approach to signal recovery which we term $b$-bit Marginal Regression. This approach builds on a method for one-bit compressed sensing proposed in an influential paper by Plan and Vershynin [23] which has subsequently been refined in several recent works [4, 24, 28]. As indicated by the name, $b$-bit Marginal Regression can be seen as a quantized version of Marginal Regression, a simple

yet surprisingly effective approach to support recovery that stands out due to its low computational cost, requiring only a single matrix-vector multiplication and a sorting operation [7]. Our analysis yields a precise characterization of the above trade-off involving $m$ and $b$ in various settings. It turns out that the choice $b = 1$ is optimal for recovering the normalized signal $x_u^* = x^*/\|x^*\|_2$, under additive Gaussian noise as well as under adversarial noise. It is shown that the choice $b = 2$ additionally enables one to estimate $\|x^*\|_2$, while being optimal for recovering $x_u^*$ for $b \geq 2$. Hence for the specific recovery algorithm under consideration, it does not pay off to take $b > 2$. Furthermore, once the noise level is significantly high, $b$-bit Marginal Regression is empirically shown to perform roughly as good as several alternative recovery algorithms, a finding suggesting that in high-noise settings taking $b > 2$ does not pay off in general. As an intermediate step in our analysis, we prove that Lloyd-Max quantization [19, 20] constitutes an optimal $b$-bit quantization scheme in the sense that it leads to a minimization of an upper bound on the reconstruction error.

**Notation**: We use $[d] = \{1, \ldots, d\}$ and $S(x)$ for the support of $x \in \mathbb{R}^n$. $x \odot x' = (x_j \cdot x_j')_{j=1}^n$. $I(P)$ is the indicator function of expression $P$. The symbol $\propto$ means "up to a positive universal constant". **Supplement**: Proofs and additional experiments can be found in the supplement.

## 2 From Marginal Regression to $b$-bit Marginal Regression

**Some background on Marginal Regression.** It is common to perform sparse signal recovery by solving an optimization problem of the form

$$\min_x \frac{1}{2m}\|y - Ax\|_2^2 + \frac{\gamma}{2}P(x), \;\; \gamma \geq 0, \tag{2}$$

where $P$ is a penalty term encouraging sparse solutions. Standard choices for $P$ are $P(x) = \|x\|_0$, which is computationally not feasible in general, its convex relaxation $P(x) = \|x\|_1$ or non-convex penalty terms like SCAD or MCP that are more amenable to optimization than the $\ell_0$-norm [27]. Alternatively $P$ can as well be used to enforce a constraint by setting $P(x) = \iota_\mathcal{C}(x)$, where $\iota_\mathcal{C}(x) = 0$ if $x \in \mathcal{C}$ and $+\infty$ otherwise, with $\mathcal{C} = \{x \in \mathbb{R}^n : \|x\|_0 \leq s\}$ or $\mathcal{C} = \{x \in \mathbb{R}^n : \|x\|_1 \leq r\}$ being standard choices. Note that (2) is equivalent to the optimization problem

$$\min_x -\langle \eta, x \rangle + \frac{1}{2}x^\top \frac{A^\top A}{m} x + \frac{\gamma}{2}P(x), \quad \text{where } \eta = \frac{A^\top y}{m}.$$

Replacing $A^\top A/m$ by $\mathbf{E}[A^\top A/m] = I$ (recall that the entries of $A$ are i.i.d. $N(0,1)$), we obtain

$$\min_x -\langle \eta, x \rangle + \frac{1}{2}\|x\|_2^2 + \frac{\gamma}{2}P(x), \;\; \eta = \frac{A^\top y}{m}, \tag{3}$$

which tends to be much simpler to solve than (2) as the first two terms are separable in the components of $x$. For the choices of $P$ mentioned above, we obtain closed form solutions:

$$P(x) = \|x\|_0 : \widehat{x}_j = \eta_j I(|\eta_j| \geq \gamma^{1/2}) \qquad P(x) = \|x\|_1 : \widehat{x}_j = (|\eta_j| - \gamma)_+ \operatorname{sign}(\eta_j),$$
$$P(x) = \iota_{x:\|x\|_0 \leq s} : \widehat{x}_j = \eta_j I(|\eta_j| \geq |\eta_{(s)}|) \;\; P(x) = \iota_{x:\|x\|_1 \leq r} : \widehat{x}_j = (|\eta_j| - \gamma^*)_+ \operatorname{sign}(\eta_j) \tag{4}$$

for $j \in [n]$, where $_+$ denotes the positive part and $|\eta_{(s)}|$ is the $s$th largest entry in $\eta$ in absolute magnitude and $\gamma^* = \min\{\gamma \geq 0 : \sum_{j=1}^n (|\eta_j| - \gamma)_+ \leq r\}$. In other words, the estimators are hard-respectively soft-thresholded versions of $\eta_j = A_j^\top y/m$ which are essentially equal to the univariate (or marginal) regression coefficients $\theta_j = A_j^\top y/\|A_j\|_2^2$ in the sense that $\eta_j = \theta_j(1 + O_\mathbf{P}(m^{-1}))$, $j \in [n]$, hence the term "marginal regression". In the literature, it is the estimator in the left half of (4) that is popular [7], albeit as a means to infer the support of $x^*$ rather than $x^*$ itself. Under (2) the performance with respect to signal recovery can still be reasonable in view of the statement below.

**Proposition 1.** *Consider model* (1) *with* $x^* \neq 0$ *and the Marginal Regression estimator* $\widehat{x}$ *defined component-wise by* $\widehat{x}_j = \eta_j I(|\eta_j| \geq |\eta_{(s)}|)$, $j \in [n]$, *where* $\eta = A^\top y/m$. *Then there exists positive constants* $c, C > 0$ *such that with probability at least* $1 - cn^{-1}$

$$\frac{\|\widehat{x} - x^*\|_2}{\|x^*\|_2} \leq C \frac{\|x^*\|_2 + \sigma}{\|x^*\|_2} \sqrt{\frac{s \log n}{m}}. \tag{5}$$

In comparison, the relative $\ell_2$-error of more sophisticated methods like the lasso scales as $O(\{\sigma/\|x^*\|_2\} \sqrt{s \log(n)/m})$ which is comparable to (5) once $\sigma$ is of the same order of magnitude as $\|x^*\|_2$. Marginal Regression can also be interpreted as a single projected gradient iteration

from 0 for problem (2) with $P = \iota_{x:\|x\|_0 \leq s}$. Taking more than one projected gradient iteration gives rise to a popular recovery algorithm known as Iterative Hard Thresholding (IHT, [1]).

**Compressed sensing with non-linear observations and the method of Plan & Vershynin.** As a generalization of (1) one can consider measurements of the form

$$y_i = Q(\langle a_i, x^* \rangle + \sigma \varepsilon_i), \quad i \in [m] \tag{6}$$

for some map $Q$. Without loss generality, one may assume that $\|x^*\|_2 = 1$ as long as $x^* \neq 0$ (which is assumed in the sequel) by defining $Q$ accordingly. Plan and Vershynin [23] consider the following optimization problem for recovering $x^*$, and develop a framework for analysis that covers even more general measurement models than (6). The proposed estimator minimizes

$$\min_{x:\|x\|_2 \leq 1, \|x\|_1 \leq \sqrt{s}} - \langle \eta, x \rangle, \quad \eta = A^\top y/m. \tag{7}$$

Note that the constraint set $\{x : \|x\|_2 \leq 1, \|x\|_1 \leq \sqrt{s}\}$ contains $\{x : \|x\|_2 \leq 1, \|x\|_0 \leq s\}$. The authors prefer the former because it is suited for approximately sparse signals as well and second because it is convex. However, the optimization problem with sparsity constraint is easy to solve:

$$\min_{x:\|x\|_2 \leq 1, \|x\|_0 \leq s} - \langle \eta, x \rangle, \quad \eta = A^\top y/m. \tag{8}$$

**Lemma 1.** *The solution of problem* (8) *is given by* $\widehat{x} = \widetilde{x}/\|\widetilde{x}\|_2, \; \widetilde{x}_j = \eta_j I(|\eta_j| \geq |\eta_{(s)}|), \; j \in [n].$

While this is elementary we state it as a separate lemma as there has been some confusion in the existing literature. In [4] the same solution is obtained after (unnecessarily) convexifying the constraint set, which yields the unit ball of the so-called $s$-support norm. In [24] a family of concave penalty terms including the SCAD and MCP is proposed in place of the cardinality constraint. However, in light of Lemma 1, the use of such penalty terms lacks motivation.

The minimization problem (8) is essentially that of Marginal Regression (3) with $P = \iota_{x:\|x\|_0 \leq s}$, the only difference being that the norm of the solution is fixed to one. Note that the Marginal Regression estimator is equi-variant w.r.t. re-scaling of $y$, i.e. for $a \cdot y$ with $a > 0$, $\widehat{x}$ changes to $a\widehat{x}$. In addition, let $\alpha, \beta > 0$ and define $\widehat{x}(\alpha)$ and $\widehat{x}[\beta]$ as the minimizers of the optimization problems

$$\min_{x:\|x\|_0 \leq s} - \langle \eta, x \rangle + \frac{\alpha}{2}\|x\|_2^2, \quad \min_{x:\|x\|_2 \leq \beta, \|x\|_0 \leq s} - \langle \eta, x \rangle. \tag{9}$$

It is not hard to verify that $\widehat{x}(\alpha)/\|\widehat{x}(\alpha)\|_2 = \widehat{x}[\beta]/\|\widehat{x}[\beta]\|_2 = \widehat{x}[1]$. In summary, for estimating the direction $x_u^* = x^*/\|x^*\|_2$ it does not matter if a quadratic term in the objective or an $\ell_2$-norm constraint is used. Moreover, estimation of the 'scale' $\psi^* = \|x^*\|_2$ and the direction can be separated. Adopting the framework in [23], we provide a straightforward bound on the $\ell_2$-error of $\widehat{x}$ minimizing (8). To this end we define two quantities which will be of central interest in subsequent analysis.

$$\lambda = \mathbf{E}[g\,\theta(g)], \; g \sim N(0,1), \quad \text{where } \theta \text{ is defined by } \mathbf{E}[y_1|a_1] = \theta(\langle a_1, x^* \rangle)$$

$$\Psi = \inf\{C > 0 : \mathbf{P}\{\max_{1 \leq j \leq n} |\eta_j - \mathbf{E}[\eta_j]| \leq C\sqrt{\log(n)/m}\} \geq 1 - 1/n.\}. \tag{10}$$

The quantity $\lambda$ concerns the deterministic part of the analysis as it quantifies the distortion of the linear measurements under the map $Q$, while $\Psi$ is used to deal with the stochastic part. The definition of $\Psi$ is based on the usual tail bound for the maximum of centered sub-Gaussian random variables. In fact, as long as $Q$ has bounded range, Gaussianity of the $\{A_{ij}\}$ implies that the $\{\eta_j - \mathbf{E}[\eta_j]\}_{j=1}^n$ are zero-mean sub-Gaussian. Accordingly, the constant $\Psi$ is proportional to the sub-Gaussian norm of the $\{\eta_j - \mathbf{E}[\eta_j]\}_{j=1}^n$, cf. [25].

**Proposition 2.** *Consider model* (6) *s.t.* $\|x^*\|_2 = 1$ *and* (10). *Suppose that* $\lambda > 0$ *and denote by* $\widehat{x}$ *the minimizer of* (8). *Then with probability at least* $1 - 1/n$, *it holds that*

$$\|x^* - \widehat{x}\|_2 \leq 2\sqrt{2}\,\frac{\Psi}{\lambda}\sqrt{\frac{s\log n}{m}}. \tag{11}$$

So far $s$ has been assumed to be known. If that is not the case, $s$ can be estimated as follows.

**Proposition 3.** *In the setting of Proposition 2, consider* $\widehat{s} = |\{j : |\eta_j| > \Psi\sqrt{\log(n)/m}\}|$ *and* $\widehat{x}$ *as the minimizer of* (8) *with* $s$ *replaced by* $\widehat{s}$. *Then with probability at least* $1 - 1/n$, $S(\widehat{x}) \subseteq S(x^*)$ *(i.e. no false positive selection). Moreover, if*

$$\min_{j \in S(x^*)} |x_j^*| > (2\Psi/\lambda)\sqrt{\log(n)/m}, \; \text{one has } S(\widehat{x}) = S(x^*). \tag{12}$$

$b$-**bit Marginal Regression.** $b$-bit quantized measurements directly fit into the non-linear observation model (6). Here the map $Q$ represents a quantizer that partitions $\mathbb{R}_+$ into $K = 2^{b-1}$ bins $\{\mathcal{R}_k\}_{k=1}^{K}$ given by distinct thresholds $\mathbf{t} = (t_1, \ldots, t_{K-1})^\top$ (in increasing order) and $t_0 = 0$, $t_K = +\infty$ such that $\mathcal{R}_1 = [t_0, t_1), \ldots, \mathcal{R}_K = [t_{K-1}, t_K)$. Each bin is assigned a distinct representative from $\mathcal{M} = \{\mu_1, \ldots, \mu_K\}$ (in increasing order) so that $Q : \mathbb{R} \rightarrow -\mathcal{M} \cup \mathcal{M}$ is defined by $z \mapsto Q(z) = \text{sign}(z) \sum_{k=1}^{K} \mu_k I(|z| \in \mathcal{R}_k)$. Expanding model (6) accordingly, we obtain

$$y_i = \text{sign}(\langle a_i, x^* \rangle + \sigma \varepsilon_i) \sum_{k=1}^{K} \mu_k I(|(\langle a_i, x^* \rangle + \sigma \varepsilon_i)| \in \mathcal{R}_k)$$
$$= \text{sign}(\langle a_i, x_u^* \rangle + \tau \varepsilon_i) \sum_{k=1}^{K} \mu_k I(|(\langle a_i, x_u^* \rangle + \tau \varepsilon_i)| \in \mathcal{R}_k/\psi^*), \; i \in [m],$$

where $\psi^* = \|x^*\|_2$, $x_u^* = x^*/\psi^*$ and $\tau = \sigma/\psi^*$. Thus the scale $\psi^*$ of the signal can be absorbed into the definition of the bins respectively thresholds which should be proportional to $\psi^*$. We may thus again fix $\psi^* = 1$ and in turn $x^* = x_u^*$, $\sigma = \tau$ w.l.o.g. for the analysis below. Estimation of $\psi^*$ separately from $x_u^*$ will be discussed in an extra section.

## 3 Analysis

In this section we study in detail the central question of the introduction. Suppose we have a fixed budget $B$ of bits available and are free to choose the number of measurements $m$ and the number of bits per measurement $b$ subject to $B = m \cdot b$ such that the $\ell_2$-error $\|\widehat{x} - x^*\|_2$ of $b$-bit Marginal Regression is as small as possible. What is the optimal choice of $(m, b)$? In order to answer this question, let us go back to the error bound (11). That bound applies to $b$-bit Marginal Regression for any choice of $b$ and varies with $\lambda = \lambda_b$ and $\Psi = \Psi_b$, both of which additionally depend on $\sigma$, the choice of the thresholds $\mathbf{t}$ and the representatives $\boldsymbol{\mu}$. It can be shown that the dependence of (11) on the ratio $\Psi/\lambda$ is tight asymptotically as $m \rightarrow \infty$. Hence it makes sense to compare two different choices $b$ and $b'$ in terms of the ratio of $\Omega_b = \Psi_b/\lambda_b$ and $\Omega_{b'} = \Psi_{b'}/\lambda_{b'}$. Since the bound (11) decays with $\sqrt{m}$, for $b'$-bit measurements, $b' > b$, to improve over $b$-bit measurements with respect to the total #bits used, it is then required that $\Omega_b/\Omega_{b'} > \sqrt{b'/b}$. The route to be taken is thus as follows: we first derive expressions for $\lambda_b$ and $\Psi_b$ and then minimize the resulting expression for $\Omega_b$ w.r.t. the free parameters $\mathbf{t}$ and $\boldsymbol{\mu}$. We are then in position to compare $\Omega_b/\Omega_{b'}$ for $b \neq b'$.

**Evaluating $\lambda_b = \lambda_b(\mathbf{t}, \boldsymbol{\mu})$.** Below, $\odot$ denotes the entry-wise multiplication between vectors.

**Lemma 2.** We have $\lambda_b(\mathbf{t}, \boldsymbol{\mu}) = \langle \boldsymbol{\alpha}(\mathbf{t}), \boldsymbol{E}(\mathbf{t}) \odot \boldsymbol{\mu} \rangle / (1 + \sigma^2)$, where

$$\boldsymbol{\alpha}(\mathbf{t}) = (\alpha_1(\mathbf{t}), \ldots, \alpha_K(\mathbf{t}))^\top, \quad \alpha_k(\mathbf{t}) = \mathbf{P}\{|\widetilde{g}| \in \mathcal{R}_k(\mathbf{t})\}, \; \widetilde{g} \sim N(0, 1 + \sigma^2), \; k \in [K],$$
$$\boldsymbol{E}(\mathbf{t}) = (E_1(\mathbf{t}), \ldots, E_K(\mathbf{t}))^\top, \quad E_k(\mathbf{t}) = \mathbf{E}[\widetilde{g}|\widetilde{g} \in \mathcal{R}_k(\mathbf{t})], \quad \widetilde{g} \sim N(0, 1 + \sigma^2), \; k \in [K].$$

**Evaluating $\Psi_b = \Psi_b(\mathbf{t}, \boldsymbol{\mu})$.** Exact evaluation proves to be difficult. We hence resort to an analytically more tractable approximation which is still sufficiently accurate as confirmed by experiments.

**Lemma 3.** As $|x_j^*| \rightarrow 0$, $j = 1, \ldots, n$, and as $m \rightarrow \infty$, we have $\Psi_b(\mathbf{t}, \boldsymbol{\mu}) \propto \sqrt{\langle \boldsymbol{\alpha}(\mathbf{t}), \boldsymbol{\mu} \odot \boldsymbol{\mu} \rangle}$.

Note that the proportionality constant (not depending on $b$) in front of the given expression does not need to be known as it cancels out when computing ratios $\Omega_b/\Omega_{b'}$. The asymptotics $|x_j^*| \rightarrow 0$, $j \in [n]$, is limiting but still makes sense for $s$ growing with $n$ (recall that we fix $\|x^*\|_2 = 1$ w.l.o.g.).

**Optimal choice of $\mathbf{t}$ and $\boldsymbol{\mu}$.** It turns that the optimal choice of $(\mathbf{t}, \boldsymbol{\mu})$ minimizing $\Psi_b/\lambda_b$ coincides with the solution of an instance of the classical Lloyd-Max quantization problem [19, 20] stated below. Let $h$ be a random variable with finite variance and $Q$ the quantization map from above.

$$\min_{\mathbf{t}, \boldsymbol{\mu}} \mathbf{E}[\{h - Q(h; \mathbf{t}, \boldsymbol{\mu})\}^2] = \min_{\mathbf{t}, \boldsymbol{\mu}} \mathbf{E}[\{h - \text{sign}(h) \sum_{k=1}^{K} \mu_k I(|h| \in \mathcal{R}_k(\mathbf{t}))\}^2]. \quad (13)$$

Problem (13) can be seen as a one-dimensional $k$-means problem at the population level, and it is solved in practice by an alternating scheme similar to that used for $k$-means. For $h$ from a log-concave distribution (e.g. Gaussian) that scheme can be shown to deliver the global optimum [12].

**Theorem 1.** Consider the minimization problem $\min_{\mathbf{t}, \boldsymbol{\mu}} \Psi_b(\mathbf{t}, \boldsymbol{\mu})/\lambda_b(\mathbf{t}, \boldsymbol{\mu})$. Its minimizer $(\mathbf{t}^*, \boldsymbol{\mu}^*)$ equals that of the Lloyd-Max problem (13) for $h \sim N(0, 1 + \sigma^2)$. Moreover,

$$\Omega_b(\mathbf{t}^*, \boldsymbol{\mu}^*) = \Psi_b(\mathbf{t}^*, \boldsymbol{\mu}^*)/\lambda_b(\mathbf{t}^*, \boldsymbol{\mu}^*) \propto \sqrt{(\sigma^2 + 1)/\lambda_{b,0}(\boldsymbol{t}_0^*, \boldsymbol{\mu}_0^*)},$$

where $\lambda_{b,0}(\boldsymbol{t}_0^*, \boldsymbol{\mu}_0^*)$ denotes the value of $\lambda_b$ for $\sigma = 0$ evaluated at $(\mathbf{t}_0^*, \boldsymbol{\mu}_0^*)$, the choice of $(\mathbf{t}, \boldsymbol{\mu})$ minimizing $\Omega_b$ for $\sigma = 0$.

Regarding the choice of $(\mathbf{t}, \boldsymbol{\mu})$ the result of Theorem 1 may not come as a suprise as the entries of $y$ are i.i.d. $N(0, 1 + \sigma^2)$. It is less immediate though that this specific choice can also be motivated as the one leading to the minimization of the error bound (11). Furthermore, Theorem 1 implies that the relative performance of $b$- and $b'$-bit measurements does not depend on $\sigma$ as long as the respective optimal choice of $(\mathbf{t}, \boldsymbol{\mu})$ is used, which requires $\sigma$ to be known. Theorem 1 provides an explicit expression for $\Omega_b$ that is straightforward to compute. The following table lists ratios $\Omega_b/\Omega_{b'}$ for selected values of $b$ and $b'$.

|  | $b=1, b'=2$ | $b=2, b'=3$ | $b=3, b'=4$ |
|---|---|---|---|
| $\Omega_b/\Omega_{b'}$: | **1.178** | **1.046** | **1.013** |
| required for $b' \gg b$: | $\sqrt{2} \approx \mathbf{1.414}$ | $\sqrt{3/2} \approx \mathbf{1.225}$ | $\sqrt{4/3} \approx \mathbf{1.155}$ |

These figures suggests that the smaller $b$, the better the performance for a given budget of bits $B$.

**Beyond additive noise.** Additive Gaussian noise is perhaps the most studied form of perturbation, but one can of course think of numerous other mechanisms whose effect can be analyzed on the basis of the same scheme used for additive noise as long as it is feasible to obtain the corresponding expressions for $\lambda$ and $\Psi$. We here do so for the following mechanisms acting *after* quantization.

(I) *Random bin flip.* For $i \in [m]$: with probability $1 - p$, $y_i$ remains unchanged. With probability $p$, $y_i$ is changed to an element from $(-\mathcal{M} \cup \mathcal{M}) \setminus \{y_i\}$ uniformly at random.

(II) *Adversarial bin flip.* For $i \in [m]$: Write $y_i = q\mu_k$ for $q \in \{-1, 1\}$ and $\mu_k \in \mathcal{M}$. With probability $1 - p$, $y_i$ remains unchanged. With probability $p$, $y_i$ is changed to $-q\mu_K$.

Note that for $b = 1$, (I) and (II) coincide (sign flip with probability $p$). Depending on the magnitude of $p$, the corresponding value $\lambda = \lambda_{b,p}$ may even be negative, which is unlike the case of additive noise. Recall that the error bound (11) requires $\lambda > 0$. Borrowing terminology from robust statistics, we consider $\bar{p}_b = \min\{p : \lambda_{b,p} \leq 0\}$ as the *breakdown point*, i.e. the (expected) proportion of contaminated observations that can still be tolerated so that (11) continues to hold. Mechanism (II) produces a natural counterpart of gross corruptions in the standard setting (1). It can be shown that among all maps $-\mathcal{M} \cup \mathcal{M} \to -\mathcal{M} \cup \mathcal{M}$ applied randomly to the observations with a fixed probability, (II) maximizes the ratio $\Psi/\lambda$, hence the attribute "adversarial". In Figure 1 we display $\Psi_{b,p}/\lambda_{b,p}$ for $b \in \{1, 2, 3, 4\}$ for both (I) and (II). The table below lists the corresponding breakdown points. For simplicity, $(\mathbf{t}, \boldsymbol{\mu})$ are not optimized but set to the optimal (in the sense of Lloyd-Max) choice $(\mathbf{t}_0^*, \boldsymbol{\mu}_0^*)$ in the noiseless case. The underlying derivations can be found in the supplement.

| (I) | $b=1$ | $b=2$ | $b=3$ | $b=4$ | (II) | $b=1$ | $b=2$ | $b=3$ | $b=4$ |
|---|---|---|---|---|---|---|---|---|---|
| $\bar{p}_b$ | $1/2$ | $3/4$ | $7/8$ | $15/16$ | $\bar{p}_b$ | $1/2$ | $0.42$ | $0.36$ | $0.31$ |

Figure 1 and the table provide one more argument in favour of one-bit measurements as they offer better robustness vis-à-vis adversarial corruptions. In fact, once the fraction of such corruptions reaches $0.2$, $b = 1$ performs best − on the measurement scale. For the milder corruption scheme (I), $b = 2$ turns out to the best choice for significant but moderate $p$.

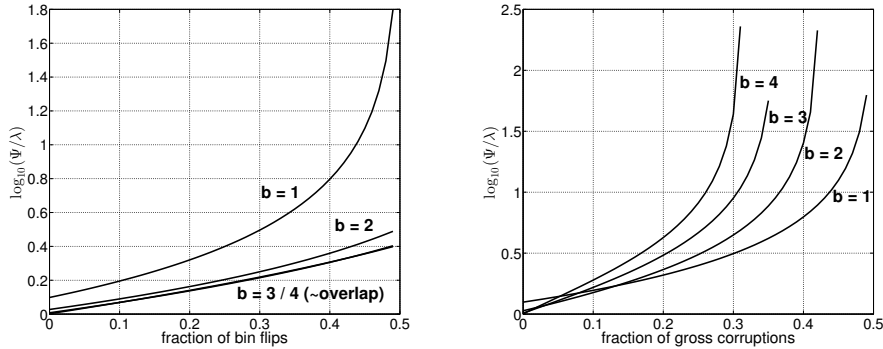

Figure 1: $\Psi_{b,p}/\lambda_{b,p}$ ($\log_{10}$-scale), $b \in \{1, 2, 3, 4\}$, $p \in [0, 0.5]$ for mechanisms (I, **L**) and (II, **R**).

## 4 Scale estimation

In Section 2, we have decomposed $x^* = x_u^* \psi^*$ into a product of a unit vector $x_u^*$ and a scale parameter $\psi^* > 0$. We have pointed out that $x_u^*$ can be estimated by $b$-bit Marginal Regression

separately from $\psi^*$ since the latter can be absorbed into the definition of the bins $\{\mathcal{R}_k\}$. Accordingly, we may estimate $x^*$ as $\widehat{x} = \widehat{x}_u \widehat{\psi}$ with $\widehat{x}_u$ and $\widehat{\psi}$ estimating $x_u^*$ and $\psi^*$, respectively. We here consider the maximum likelihood estimator (MLE) for $\psi^*$, by following [15] which studied the estimation of the scale parameter for the entire $\alpha$-stable family of distributions. The work of [15] was motivated by a different line of one scan 1-bit CS algorithm [16] based on $\alpha$-stable designs [17].

First, we consider the case $\sigma = 0$, so that the $\{y_i\}$ are i.i.d. $N(0, (\psi^*)^2)$. The likelihood function is

$$L(\psi) = \prod_{i=1}^{m} \sum_{k=1}^{K} I(y_i \in \mathcal{R}_k) \, \mathbf{P}(|y_i| \in \mathcal{R}_k) = \prod_{k=1}^{K} \{2(\Phi(t_k/\psi) - \Phi(t_{k-1}/\psi))\}^{m_k}, \quad (14)$$

where $m_k = |\{i : |y_i| \in \mathcal{R}_k\}|$, $k \in [K]$, and $\Phi$ denotes the standard Gaussian cdf. Note that for $K = 1$, $L(\psi)$ is constant (i.e. does not depend on $\psi$) which confirms that for $b = 1$, it is impossible to recover $\psi^*$. For $K = 2$ (i.e. $b = 2$), the MLE has a simple a closed form expression given by $\widehat{\psi} = t_1/\Phi^{-1}(0.5(1 + m_1/m))$. The following tail bound establishes fast convergence of $\widehat{\psi}$ to $\psi^*$.

**Proposition 4.** *Let $\varepsilon \in (0, 1)$ and $c = 2\{\phi'(t_1/\psi^*)\}^2$, where $\phi'$ denotes the derivative of the standard Gaussian pdf. With probability at least $1 - 2\exp(-cm\varepsilon^2)$, we have $|\widehat{\psi}/\psi^* - 1| \le \varepsilon$.*

The exponent $c$ is maximized for $t_1 = \psi^*$ and becomes smaller as $t_1/\psi^*$ moves away from 1. While scale estimation from 2-bit measurements is possible, convergence can be slow if $t_1$ is not well chosen. For $b \ge 3$, convergence can be faster but the MLE is not available in closed form [15].

We now turn to the case $\sigma > 0$. The MLE based on (14) is no longer consistent. If $x_u^*$ is known then the joint likelihood of for $(\psi^*, \sigma)$ is given by

$$L(\psi, \widetilde{\sigma}) = \prod_{i=1}^{m} \left\{ \Phi\left( \frac{u_i - \psi \langle a_i, x_u^* \rangle}{\widetilde{\sigma}} \right) - \Phi\left( \frac{l_i - \psi \langle a_i, x_u^* \rangle}{\widetilde{\sigma}} \right) \right\}, \quad (15)$$

where $[l_i, u_i]$ denotes the interval the $i$-th observation is contained in before quantization, $i \in [m]$. It is not clear to us whether the likelihood is log-concave, which would ensure that the global optimum can be obtained by convex programming. Empirically, we have not encountered any issue with spurious local minima when using $\psi = 0$ and $\widetilde{\sigma}$ as the MLE from the noiseless case as starting point. The only issue with (15) we are aware of concerns the case in which there exists $\psi$ so that $\psi \langle a_i, x_u^* \rangle \in [l_i, u_i], i \in [m]$. In this situation, the MLE for $\sigma$ equals zero and the MLE for $\psi$ may not be unique. However, this is a rather unlikely scenario as long as there is a noticeable noise level. As $x_u^*$ is typically unknown, we may follow the plug-in principle, replacing $x_u^*$ by an estimator $\widehat{x}_u$.

## 5 Experiments

We here provide numerical results supporting/illustrating some of the key points made in the previous sections. We also compare $b$-bit Marginal Regression to alternative recovery algorithms.

**Setup.** Our simulations follow model (1) with $n = 500$, $s \in \{10, 20, \ldots, 50\}$, $\sigma \in \{0, 1, 2\}$ and $b \in \{1, 2\}$. Regarding $x^*$, the support and its signs are selected uniformly at random, while the absolute magnitude of the entries corresponding to the support are drawn from the uniform distribution on $[\beta, 2\beta]$, where $\beta = f \cdot (1/\lambda_{1,\sigma})\sqrt{\log(n)/m}$ and $m = f^2(1/\lambda_{1,\sigma})^2 s \log n$ with $f \in \{1.5, 3, 4.5, \ldots, 12\}$ controlling the signal strength. The resulting signal is then normalized to unit 2-norm. Before normalization, the norm of the signal lies in $[1, \sqrt{2}]$ by construction which ensures that as $f$ increases the signal strength condition (12) is satisfied with increasing probability. For $b = 2$, we use Lloyd-Max quantization for a $N(0, 1)$-random variable which is optimal for $\sigma = 0$, but not for $\sigma > 0$. Each possible configuration for $s, f$ and $\sigma$ is replicated 20 times. Due to space limits, a representative subset of the results is shown; the rest can be found in the supplement.

**Empirical verification of the analysis in Section 3**. The experiments reveal that what is predicted by the analysis of the comparison of the relative performance of 1-bit and 2-bit measurements for estimating $x^*$ closely agrees with what is observed empirically, as can be seen in Figure 2.

**Estimation of the scale and the noise level.** Figure 3 suggests that the plug-in MLE for $(\psi^* = \|x^*\|_2, \sigma)$ is a suitable approach, at least as long as $\psi^*/\sigma$ is not too small. For $\sigma = 2$, the plug-in MLE for $\psi^*$ appears to have a noticeable bias as it tends to 0.92 instead of 1 for increasing $f$ (and thus increasing $m$). Observe that for $\sigma = 0$, convergence to the true value 1 is smaller as for $\sigma = 1$,

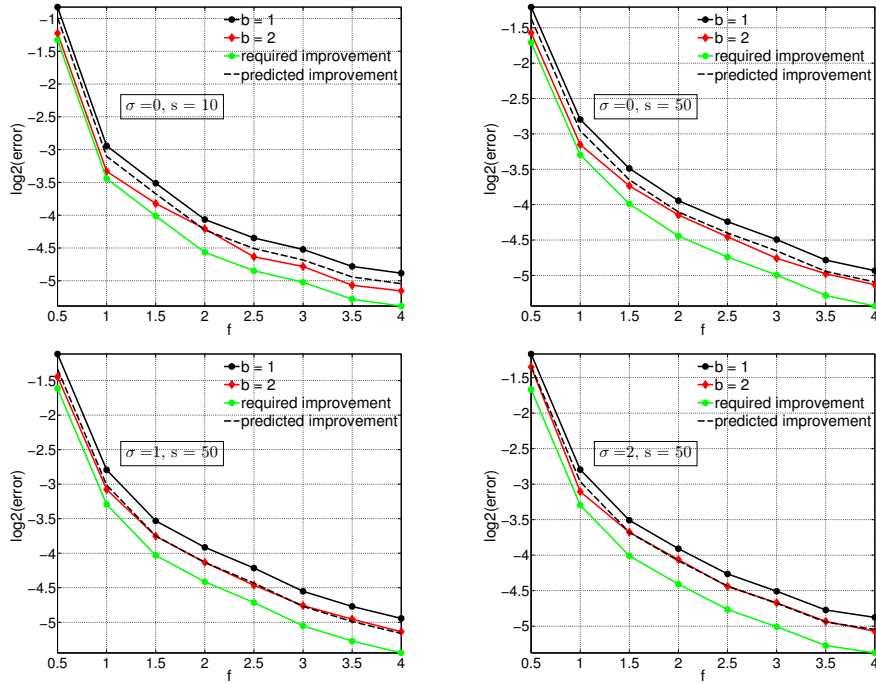

Figure 2: Average $\ell_2$-estimation errors $\|x^* - \widehat{x}\|_2$ for $b = 1$ and $b = 2$ on the $\log_2$-scale in dependence of the signal strength $f$. The curve 'predicted improvement' (of $b = 2$ vs. $b = 1$) is obtained by scaling the $\ell_2$-estimation error by the factor predicted by the theory of Section 3. Likewise the curve 'required improvement' results by scaling the error of $b = 1$ by $1/\sqrt{2}$ and indicates what would be required by $b = 2$ to improve over $b = 1$ at the level of total #bits.

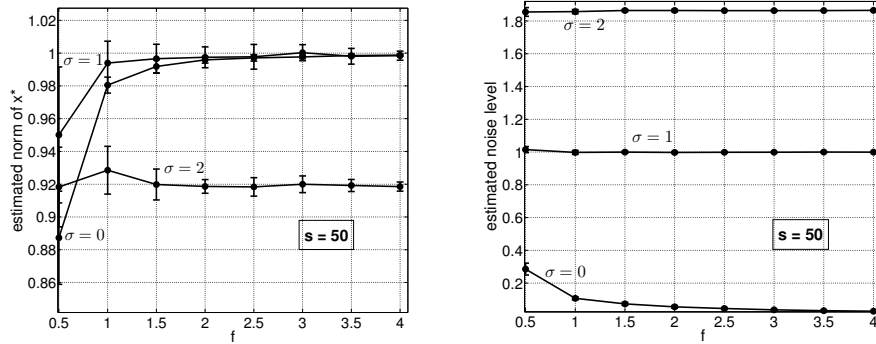

Figure 3: Estimation of $\psi = \|x^*\|_2$ (here 1) and $\sigma$. The curves depict the average of the plug-in MLE discussed in Section 4 while the bars indicate $\pm 1$ standard deviation.

while $\sigma$ is over-estimated (about $0.2$) for small $f$. The above two issues are presumably a plug-in effect, i.e. a consequence of using $\widehat{x}_u$ in place of $x_u^*$.

**$b$-bit Marginal Regression and alternative recovery algorithms.** We compare the $\ell_2$-estimation error of $b$-bit Marginal Regression to several common recovery algorithms. Compared to apparently more principled methods which try to enforce agreement of $Q(y)$ and $Q(A\widehat{x})$ w.r.t. the Hamming distance (or a surrogate thereof), $b$-bit Marginal Regression can be seen as a crude approach as it is based on maximizing the inner product between $y$ and $Ax$. One may thus expect that its performance is inferior. In summary, our experiments confirm that this is true in low-noise settings, but not so if the noise level is substantial. Below we briefly present the alternatives that we consider.

Plan-Vershynin: The approach in [23] based on (7) which only differs in that the constraint set results from a relaxation. As shown in Figure 4 the performance is similar though slightly inferior.

IHT-quadratic: Standard Iterative Hard Thresholding based on quadratic loss [1]. As pointed out above, $b$-bit Marginal Regression can be seen as one-step version of Iterative Hard Thresholding.

**IHT-hinge** ($b = 1$): The variant of Iterative Hard Threshold for binary observations using a hinge loss-type loss function as proposed in [11].

**SVM** ($b = 1$): Linear SVM with squared hinge loss and an $\ell_1$-penalty, implemented in LIBLINEAR [6]. The cost parameter is chosen from $1/\sqrt{m \log m} \cdot \{2^{-3}, 2^{-2}, \ldots, 2^3\}$ by 5-fold cross-validation.

**IHT-Jacques** ($b = 2$): A variant of Iterative Hard Threshold for quantized observations based on a specific piecewiese linear loss function [9].

**SVM-type** ($b = 2$): This approach is based on solving the following convex optimization problem: $\min_{x,\{\xi_i\}} \gamma \|x\|_1 + \sum_{i=1}^m \xi_i$ subject to $l_i - \xi_i \leq \langle a_i, x \rangle \leq u_i + \xi_i$, $\xi_i \geq 0$, $i \in [m]$, where $[l_i, u_i]$ is the bin observation $i$ is assigned to. The essential idea is to enforce consistency of the observed and predicted bin assignments up to slacks $\{\xi_i\}$ while promoting sparsity of the solution via an $\ell_1$-penalty. The parameter $\gamma$ is chosen from $\sqrt{m \log m} \cdot \{2^{-10}, 2^{-9}, \ldots, 2^3\}$ by 5-fold cross-validation.

Turning to the results as depicted by Figure 4, the difference between a noiseless ($\sigma = 0$) and heavily noisy setting ($\sigma = 2$) is perhaps most striking.

$\sigma = 0$: both IHT variants significantly outperform $b$-bit Marginal Regression. By comparing errors for IHT, $b = 2$ can be seen to improve over $b = 1$ at the level of the total # bits.

$\sigma = 2$: $b$-bit Marginal Regression is on par with the best performing methods. IHT-quadratic for $b = 2$ only achieves a moderate reduction in error over $b = 1$, while IHT-hinge is supposedly affected by convergence issues. Overall, the results suggest that a setting with substantial noise favours a crude approach (low-bit measurements and conceptually simple recovery algorithms).

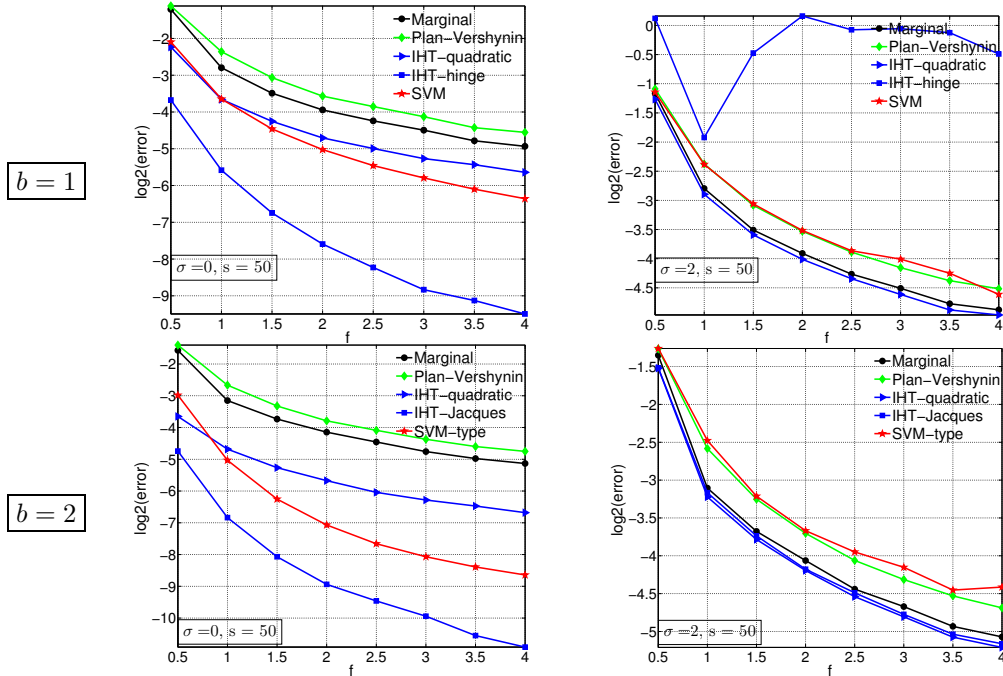

Figure 4: Average $\ell_2$-estimation errors for several recovery algorithms on the $\log_2$-scale in dependence of the signal strength $f$. We contrast $\sigma = 0$ (**L**) vs. $\sigma = 2$ (**R**), $b = 1$ (**T**) vs. $b = 2$ (**B**).

## 6 Conclusion

Bridging Marginal Regression and a popular approach to 1-bit CS due to Plan & Vershynin, we have considered signal recovery from $b$-bit quantized measurements. The main finding is that for $b$-bit Marginal Regression it is not beneficial to increase $b$ beyond 2. A compelling argument for $b = 2$ is the fact that the norm of the signal can be estimated unlike the case $b = 1$. Compared to high-precision measurements, 2-bit measurements also exhibit strong robustness properties. It is of interest if and under what circumstances the conclusion may differ for other recovery algorithms.

**Acknowledgement.** This work is partially supported by NSF-Bigdata-1419210, NSF-III-1360971, ONR-N00014-13-1-0764, and AFOSR-FA9550-13-1-0137.

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
