[Supplementary Material 1 · proofs_rev.pdf]

# Supplement to '$b$-bit Marginal Regression': proofs and derivations

## A  Proof of Proposition 1

The proof relies on concentration properties of $\chi^2$-random variables which can be found in [4], Section J.

**Lemma A.1.** *Let $Z \sim \chi^2(d)$. Then we have for any $t \in (0, 1/2)$*

$$\mathbf{P}(Z \geq d(1 + t)) \leq \exp\left(-\frac{3}{16}dt^2\right),$$

$$\mathbf{P}(Z \leq d(1 - t)) \leq \exp\left(-\frac{1}{4}dt^2\right).$$

*Proof.* (**Proposition 1**) Note that $\mathbf{E}[\eta] = \mathbf{E}[A^\top y/m] = \mathbf{E}[A^\top(Ax^* + \varepsilon)/m] = x^*$, where the expectation is w.r.t. both $A$ and $\varepsilon$. In the sequel, we will show that

$$\max_{1 \leq j \leq n} |\eta_j - \mathbf{E}[\eta_j]| = \max_{1 \leq j \leq n} \left| \frac{A_j^\top y}{m} - \mathbf{E}\left[\frac{A_j^\top y}{m}\right] \right| \leq C_0(\|x^*\|_2 + \sigma)\sqrt{\frac{\log n}{m}}. \quad (1)$$

with probability at least $1 - cn^{-1}$, for suitable constants $c, C_0 > 0$. This already implies the assertion of the proposition, as shown below. Denote

$$Q(x^*) = \left\{ j \in [n] : |x_j^*| > 2C_0(\|x^*\|_2 + \sigma)\sqrt{\frac{\log n}{m}} \right\} \subseteq S(x^*).$$

Note that under the event (1), $Q(x^*) \subseteq S(\widehat{x})$. Indeed, by the definition of $\widehat{x}$, its support $S(\widehat{x})$ contains the indices corresponding to the $s$ largest entries of $\eta$ (in absolute magnitude), and under event (1) it holds that $\min_{j \in Q(x^*)} |\eta_j| > \max_{j \in [n] \setminus S(x^*)} |\eta_j|$. We thus bound

$$
\begin{aligned}
\|\widehat{x} - x^*\|_\infty &= \max\{\|\widehat{x}_{Q(x^*)} - x^*_{Q(x^*)}\|_\infty, \|\widehat{x}_{S(x^*)\setminus Q(x^*)} - x^*_{S(x^*)\setminus Q(x^*)}\|_\infty, \|\widehat{x}_{[n]\setminus S(x^*)}\|_\infty\} \\
&\leq \max\{\|\eta_{Q(x^*)} - \mathbf{E}[\eta_{Q(x^*)}]\|_\infty, \|\eta_{S(x^*)\setminus Q(x^*)} - \mathbf{E}[\eta_{S(x^*)\setminus Q(x^*)}]\|_\infty, \\
&\qquad \|x^*_{S(x^*)\setminus Q(x^*)}\|_\infty, \|\eta_{[n]\setminus S(x^*)}\|_\infty\} \\
&\overset{(1)}{\leq} 2C_0(\|x^*\|_2 + \sigma)\sqrt{\frac{\log n}{m}}.
\end{aligned}
$$

To conclude that this yields the assertion of the proposition with $C = 2\sqrt{2}C_0$, we use

$$\|\widehat{x} - x^*\|_2 \leq \sqrt{\|\widehat{x} - x^*\|_0} \|\widehat{x} - x^*\|_\infty \leq \sqrt{2s}\|\widehat{x} - x^*\|_\infty.$$

The bound (1) can be established by standard concentration arguments. Applying the second result from Lemma A.1 with $d = m$, $t = \sqrt{8\log(n)/m}$, and using a union bound, we obtain under the assumption that $m \geq 32\log n$

$$\mathbf{P}\left(\min_{1 \leq j \leq n} \|A_j\|_2^2 \leq m - \sqrt{m}\sqrt{8\log(n)}\right) \leq \frac{1}{n}$$

$$\iff \mathbf{P}\left(\min_{1 \leq j \leq n} \|A_j\|_2^2/m \leq 1 - \sqrt{8\log(n)/m}\right) \leq \frac{1}{n}.$$

Similarly, invoking the first result of Lemma A.1 under the assumption that $m \geq (128/3) \log n$

$$\mathbf{P}\left(\max_{1 \leq j \leq n} \|A_j\|_2^2/m \geq 1 + \sqrt{(32/3)\log(n)/m}\right) \leq 1/n$$

$$\implies \quad \mathbf{P}\left(\max_{1 \leq j \leq n} \|A_j\|_2^2 \geq 2m\right) \leq 1/n.$$

Moreover, conditional on the event $\bigcap_{j \in [n]} \{\|A_j\|_2 \leq \kappa\}$ with $\kappa = \sqrt{2m}$, we have

$$\mathbf{P}\left(\max_{1 \leq j \leq n}\left|\frac{1}{m}\sum_{k \neq j}\langle A_j, A_k\rangle x_k^*\right| > t\|x^*\|_2\right) \leq 2n\exp(-m^2 t^2/(2\kappa^2)) \leq 2n\exp(-mt^2/4),$$

$$(2)$$

by a standard Gaussian tail bound. Hence, choosing $t = \sqrt{8\log(n)/m}$, we obtain

$$\mathbf{P}\left(\max_{1 \leq j \leq n}\left|\frac{1}{m}\sum_{k \neq j}\langle A_j, A_k\rangle x_k^*\right| > \|x^*\|_2\sqrt{8\log(n)/m}\right) \leq 2/n.$$

Altogether, we have with probability at least $1 - 4/n$

$$|A_j^\top(Ax^*)/m - x_j^*| \leq \left|\left(1 - \frac{\|A_j\|^2}{m}\right)x_j^*\right| + \left|\frac{1}{m}\sum_{k \neq j}\langle A_j, A_k\rangle x_k^*\right| \leq C_0\|x^*\|_2\sqrt{\log(n)/m},$$

simultaneously for all $j \in [n]$, thereby establishing (1) with $C_0 = \sqrt{32/3} + \sqrt{8}$ for $\sigma = 0$. The case $\sigma > 0$ follows immediately by the triangle inequality

$$|A_j^\top(Ax^* + \varepsilon)/m - x_j^*| \leq |A_j^\top(Ax^*)/m - x_j^*| + |A_j^\top\varepsilon/m|, \quad j \in [n],$$

and a concentration inequality for $\max_{1 \leq j \leq n}|A_j^\top\varepsilon/m|$ similarly to (2). $\square$

# B   Proof of Lemma 1

*Proof.* Let $\emptyset \neq S \subseteq \{1, \ldots, n\}$. Then for any unit vector $x$ supported on $S$, $\langle \eta, x\rangle \leq \|\eta_S\|_2$ which is attained by setting $x_S = \eta_S/\|\eta_S\|_2$. Consequently,

$$\min_{x:\|x\|_2 \leq 1, \|x\|_0 \leq s} -\langle \eta, x\rangle = \min_{S:|S| \leq s} -\|\eta_S\|_2.$$

The optimization problem on the right hand side can be solved by finding the index set of the $s$ largest component (in absolute magnitude) in $\eta$. This yields the claim. $\square$

# C   Proof of Proposition 2

For the next proof (and others below), we need the following Lemma

**Lemma C.1.** *For all $x \in \mathbb{R}^n$, we have $\mathbf{E}[\langle x, \eta\rangle] = \lambda\langle x, x_u^*\rangle$. In particular, by considering $x = e_j$, $j \in [n]$, where $\{e_j\}_{j=1}^n$ is the standard basis of $\mathbb{R}^n$, we have $\mathbf{E}[\eta] = \lambda x_u^*$.*

*Proof.*

$$\begin{aligned}
\mathbf{E}[\langle x, \eta \rangle] &= \mathbf{E}[\langle x, A^\top y/m \rangle] \\
&= \mathbf{E}[\langle Ax, y \rangle / m] \\
&= \mathbf{E}[\langle a_1, x \rangle \, y_1] \\
&= \mathbf{E}\,\mathbf{E}[y_1 \langle a_1, x \rangle \, | a_1] \\
&= \mathbf{E}[\theta(\langle a_1, x_u^* \rangle) \langle a_1, x \rangle] \\
&= \mathbf{E}\left[\theta(\langle a_1, x_u^* \rangle) \left\langle a_1, x^\parallel + x^\perp \right\rangle\right] \\
&= \langle x, x_u^* \rangle \, \mathbf{E}[\theta(g)g], \ g \sim N(0,1) \\
&= \lambda \langle x, x_u^* \rangle,
\end{aligned}$$

where in the third line from the bottom $x^\parallel = \langle x, x_u^* \rangle x_u^*$ and $x^\perp$ denote the orthogonal projection of $x$ on $x_u^*$ and its orthogonal complement, respectively. We then use that $\langle a_1, x^\perp \rangle$ and $\langle a_1, x_u^* \rangle$ are Gaussian and uncorrelated and hence also independent random variables. ☐

*Proof.* (**Proposition 2**) Since $\widehat{x}$ is a minimizer and $x_u^*$ is a feasible solution, we have

$$- \langle \eta, \widehat{x} \rangle \leq - \langle \eta, x_u^* \rangle.$$

After re-arranging, we obtain that

$$\langle x_u^* - \widehat{x}, \eta - \mathbf{E}[\eta] \rangle + \langle x_u^* - \widehat{x}, \mathbf{E}[\eta] \rangle \leq 0.$$

Using Hölder's inequality and Lemma C.1, this implies

$$\begin{aligned}
\langle x_u^* - \widehat{x}, \mathbf{E}[\eta] \rangle &\leq \|x_u^* - \widehat{x}\|_1 \|\eta - \mathbf{E}[\eta]\|_\infty \\
\langle x_u^* - \widehat{x}, \lambda x_u^* \rangle &\leq \sqrt{\|x_u^* - \widehat{x}\|_0} \|x_u^* - \widehat{x}\|_2 \|\eta - \mathbf{E}[\eta]\|_\infty \\
\frac{\lambda}{2}\|x_u^* - \widehat{x}\|_2^2 &\leq \sqrt{2s}\|x_u^* - \widehat{x}\|_2 \|\eta - \mathbf{E}[\eta]\|_\infty
\end{aligned}$$

For the last inequality, we have used that

$$\|x_u^* - \widehat{x}\|_2^2 \leq 2(1 - \langle \widehat{x}, x_u^* \rangle) = 2(\langle x_u^*, x_u^* \rangle - \langle \widehat{x}, x_u^* \rangle) = 2(\langle x_u^*, x_u^* - \widehat{x} \rangle),$$

because $\|x_u^*\|_2 = 1$ and $\|\widehat{x}\|_2 \leq 1$. Eventually, we obtain that

$$\|x_u^* - \widehat{x}\|_2 \leq 2\sqrt{2}\frac{\Psi}{\lambda}\sqrt{\frac{s \log n}{m}},$$

with probability at least $1 - 1/n$ by the definition of $\Psi$. ☐

# D    Proof of Proposition 3

*Proof.* Consider $\widehat{s} = |\{j : |\eta_j| > \Psi\sqrt{\log(n)/m}\}|$ and the optimization problem

$$\min_{x: \|x\|_2 \leq 1, \|x\|_0 \leq \widehat{s}} - \langle \eta, x \rangle.$$

Note that for $j \notin S(x^*)$, $\mathbf{E}[\eta_j] = 0$ in view of Lemma C.1, and further by the definition of $\Psi$, we have

$$\max_{j \notin S(x^*)} |\eta_j| \leq \Psi\sqrt{\log(n)/m}$$

with probability at least $1 - 1/n$. Conditional on this event, we therefore have $S(\widehat{x}) \subseteq S(x^*)$. Similarly we have

$$\min_{j \in S(x^*)} |\eta_j| \geq |\mathbf{E}[\eta_j]| - \Psi\sqrt{\log(n)/m} = \lambda|(x_u^*)_j| - \Psi\sqrt{\log(n)/m}.$$

Thus as long as

$$\min_{j \in S(x^*)} |(x_u^*)_j| > (2\Psi/\lambda)\sqrt{\log(n)/m}$$

it holds that $\min_{j \in S(x^*)} |\eta_j| > \Psi\sqrt{\log(n)/m}$ and consequently $S(\widehat{x}) = S(x^*)$. ☐

# E  Proof of Lemma 2

The proof of Lemma 2 requires three additional lemmas.

**Lemma E.1.** *Let $g \sim N(0,1)$ and $\zeta : \mathbb{R} \to \mathbb{R}$ be any differentiable function satisfying $|\zeta(x)x\phi(x)| \to 0$ as $x \to \infty$, where $\phi$ denotes the standard Gaussian pdf. Then $\mathbf{E}[\zeta(g)g] = \mathbf{E}[\zeta'(g)]$.*

*Proof.* Observe that $\phi'(x) = -x\phi(x)$, $x \in \mathbb{R}$. Using integration by parts we thus have

$$
\mathbf{E}[\zeta(g)g] = \int_{\mathbb{R}} x\zeta(x)\phi(x)\,dx = \{\zeta(x)(-\phi'(x))\}\Big|_{-\infty}^{\infty} + \int_{\mathbb{R}} \zeta'(x)\,\phi(x)\,dx
$$

$$
= \int_{\mathbb{R}} \zeta'(x)\,\phi(x)\,dx = \mathbf{E}[\zeta'(g)].
$$

$\square$

**Lemma E.2.** *For all $\alpha, \beta > 0$ and all $\mu, \nu \in \mathbb{R}$, one has*

$$
\int_{-\infty}^{\infty} \frac{1}{\alpha}\phi\left(\frac{x-\mu}{\alpha}\right)\frac{1}{\beta}\phi\left(\frac{x-\nu}{\beta}\right)dx = \frac{1}{\sqrt{\beta^2+\alpha^2}}\phi\left(\frac{\mu-\nu}{\sqrt{\beta^2+\alpha^2}}\right).
$$

*Proof.* Using elementary manipulations, one computes

$$
\int_{-\infty}^{\infty} \frac{1}{\alpha}\phi\left(\frac{x-\mu}{\alpha}\right)\frac{1}{\beta}\phi\left(\frac{x-\nu}{\beta}\right)dx
$$

$$
= \frac{1}{2\pi\alpha\beta}\int_{-\infty}^{\infty} \exp\left(-\frac{(\mu-x)^2}{2\alpha^2}\right)\exp\left(-\frac{(\nu-x)^2}{2\beta^2}\right)dx
$$

$$
= \frac{1}{2\pi\alpha\beta}\exp\left(-\frac{(\mu-\nu)^2}{2(\alpha^2+\beta^2)}\right)\times
$$

$$
\times \int_{-\infty}^{\infty} \exp\left(-\frac{1}{2}\left(\left(\frac{\mu}{\alpha^2}+\frac{\nu}{\beta^2}\right)\left(\frac{1}{\alpha^2}+\frac{1}{\beta^2}\right)-x\right)^2\left(\frac{1}{\alpha^2}+\frac{1}{\beta^2}\right)\right)dx
$$

$$
= \frac{1}{2\pi\alpha\beta}\sqrt{2\pi}\frac{\alpha\beta}{\sqrt{\beta^2+\alpha^2}}\exp\left(-\frac{(\mu-\nu)^2}{2(\alpha^2+\beta^2)}\right)
$$

$$
= \frac{1}{\sqrt{2\pi}}\frac{1}{\sqrt{\beta^2+\alpha^2}}\exp\left(-\frac{(\mu-\nu)^2}{2(\alpha^2+\beta^2)}\right).
$$

$\square$

**Lemma E.3.** *Let $h$ be a random variable with a $N(0,\sigma^2)$-distribution. Then for any $a, b \in \mathbb{R} \cup \{-\infty, \infty\}$, $a < b$, we have*

$$
\mathbf{E}[h|h \in (a,b)] = \sigma\frac{\phi(a/\sigma) - \phi(b/\sigma)}{\Phi(b/\sigma) - \Phi(a/\sigma)},
$$

*where $\Phi$ denotes the standard Gaussian cdf.*

*Proof.* We have

$$
\mathbf{E}[h|h \in (a,b)] = \frac{1}{\Phi(b/\sigma) - \Phi(a/\sigma)}\int_a^b \frac{x}{\sigma}\phi(x/\sigma)\,dx.
$$

Using the change of variables $z = x/\sigma$ and the fact that $\phi'(z) = -z\phi(z)$, the result follows. $\square$

Before finally turning to the proof of Lemma 2, let us recall the definition of the $b$-bit quantization map given at the end of Section 2. In that definition we have used the symmetry of the Gaussian distribution around 0 so that a partitioning of $\mathbb{R}_+$ automatically translates into a partitioning of $\mathbb{R}$. For parts of the proofs, however, it is more convenient to work with the following alternative (albeit equivalent) definition.

**Definition E.1.** *Define $\mathcal{Q}_1 = -\mathcal{R}_K$, $\mathcal{Q}_2 = -\mathcal{R}_{K-1}, \ldots, \mathcal{Q}_K = -\mathcal{R}_1$, $\mathcal{Q}_{K+k} = \mathcal{R}_k$, $k \in [K]$ and $\widetilde{\mu} = (-\mu_K, \ldots, \mu_1, \mu_1, \ldots, \mu_K)^\top$. Then an equivalent definition of the quantization map is given by $z \mapsto Q(z) = \sum_{k=1}^{2K} \widetilde{\mu}_k I(z \in \mathcal{Q}_k)$. Likewise, we define $\widetilde{\mathbf{t}} = (-t_K, -t_{K-1}, \ldots, t_0, t_1, \ldots, t_{K-1}, t_K)^\top$.*

*Proof.* **(Lemma 2)** Recall that $\lambda = \lambda_{b,\sigma} = \lambda_{b,\sigma}(\mathbf{t}, \boldsymbol{\mu})$ is defined by $\lambda = \mathbf{E}[g\,\theta(g)]$, $g \sim N(0,1)$, where the map $\theta$ is in turn defined by the relation $\mathbf{E}[y_1|a_1] = \theta(z_1)$ (here and below $z_i = \langle a_i, x^* \rangle$, $i \in [m]$). We have

$$\mathbf{E}[y_1|a_1] = \sum_{k=1}^{2^b} \widetilde{\mu}_k \, \mathbf{P}(y_1 \in \mathcal{Q}_k)$$

$$= \sum_{k=1}^{2^b} \widetilde{\mu}_k \, \mathbf{P}(z_1 + \varepsilon_1 \in \mathcal{Q}_k)$$

$$= \sum_{k=1}^{2^b} \widetilde{\mu}_k \, \mathbf{P}(z_1 + \varepsilon_1 \in (\widetilde{t}_k, \widetilde{t}_{k+1}))$$

$$= \sum_{k=1}^{2^b} \widetilde{\mu}_k \left\{ \Phi((\widetilde{t}_{k+1} - z_1)/\sigma) - \Phi((\widetilde{t}_k - z_1)/\sigma) \right\}.$$

We conclude that the map $\theta$ is defined by

$$\theta(z) = \sum_{k=1}^{2^b} \widetilde{\mu}_k \left\{ \Phi((\widetilde{t}_{k+1} - z)/\sigma) - \Phi((\widetilde{t}_k - z)/\sigma) \right\}.$$

Next we invoke Lemma E.1 which yields $\lambda = \mathbf{E}[z\theta(z)] = \mathbf{E}[\theta'(z)]$. We have

$$\theta'(z) = \sum_{k=1}^{2^b} \widetilde{\mu}_k \left\{ \frac{1}{\sigma}\phi((z - \widetilde{t}_k)/\sigma) - \frac{1}{\sigma}\phi((z - \widetilde{t}_{k+1})/\sigma) \right\}.$$

With the help of Lemma E.2, we compute

$$\mathbf{E}[\theta'(z)] = \sum_{k=1}^{2^b} \widetilde{\mu}_k \int_{\mathbb{R}} \left\{ \frac{1}{\sigma}\phi((z - \widetilde{t}_k)/\sigma) - \frac{1}{\sigma}\phi((z - \widetilde{t}_{k+1})/\sigma) \right\} \phi(z)\, dz.$$

$$= \sum_{k=1}^{2^b} \widetilde{\mu}_k \frac{1}{\sqrt{1+\sigma^2}} \left\{ \phi\left( \frac{\widetilde{t}_k}{\sqrt{1+\sigma^2}} \right) - \phi\left( \frac{\widetilde{t}_{k+1}}{\sqrt{1+\sigma^2}} \right) \right\}.$$

Applying Lemma E.3, the last expression can be rewritten as follows:

$$\sum_{k=1}^{2^b} \widetilde{\mu}_k \frac{1}{\sqrt{1+\sigma^2}} \left\{ \phi\left(\frac{\widetilde{t}_k}{\sqrt{1+\sigma^2}}\right) - \phi\left(\frac{\widetilde{t}_{k+1}}{\sqrt{1+\sigma^2}}\right) \right\}$$

$$= \sum_{k=1}^{2^b} \widetilde{\mu}_k \frac{\mathbf{E}[\widetilde{g}|\widetilde{g}\in(\widetilde{t}_k,\widetilde{t}_{k+1})]}{1+\sigma^2} \left\{ \Phi\left(\widetilde{t}_{k+1}/\sqrt{1+\sigma^2}\right) - \Phi\left(\widetilde{t}_k/\sqrt{1+\sigma^2}\right) \right\},$$

$$\widetilde{g} \sim N(0, 1+\sigma^2)$$

$$= \frac{1}{1+\sigma^2} \sum_{k=1}^{2^b} \widetilde{\mu}_k \, \mathbf{E}[\widetilde{g}|\widetilde{g}\in\mathcal{Q}_k] \, \mathbf{P}(\widetilde{g}\in\mathcal{Q}_k)$$

$$= \frac{1}{1+\sigma^2} \sum_{k=1}^{K} \mu_k \, \mathbf{E}[\widetilde{g}|\widetilde{g}\in\mathcal{R}_k] \, \mathbf{P}(|\widetilde{g}|\in\mathcal{R}_k)$$

$$= \frac{1}{1+\sigma^2} \langle \boldsymbol{\alpha}(\mathbf{t}), \boldsymbol{E}(\mathbf{t}) \odot \boldsymbol{\mu} \rangle,$$

where the penultimate line follows from the symmetry of the Gaussian distribution around zero; at this point, we convert the partitioning of $\mathbb{R}$ into $\{\mathcal{Q}_k\}_{k=1}^{2K}$ back to the partitioning of $\mathbb{R}_+$ into $\{\mathcal{R}_k\}_{k=1}^{K}$ (cf. the remark preceding Definition E.1). The last line follows by comparison with the definitions in Lemma 2. $\qquad\square$

# F    Proof of Lemma 3

*Proof.* Let us recall the definition of $\Psi = \Psi_{b,\sigma} = \Psi_{b,\sigma}(\mathbf{t}, \boldsymbol{\mu})$:

$$\Psi = \inf\{C > 0 : \mathbf{P}\{\max_{1\le j\le n} |\eta_j - \mathbf{E}[\eta_j]| \le C\sqrt{\log(n)/m}\} \ge 1 - 1/n.\}.$$

Expanding $\eta_j - \mathbf{E}[\eta_j]$, we obtain that

$$\eta_j - \mathbf{E}[\eta_j] = \frac{1}{m}\sum_{i=1}^{m}(A_{ij}y_i - \mathbf{E}[A_{ij}y_i]).$$

Note that since the $A_{ij}$ are i.i.d. $N(0,1)$ variables while the $\{y_i\}$ are bounded random variables, the $\{A_{ij}y_i - \mathbf{E}[A_{ij}y_i]\}_{i=1}^m$ are i.i.d. zero-mean sub-Gaussian random variables, $j \in [n]$, cf. e.g. [3]. By using a standard tail bound for such random variables and a union bound over $\{1,\ldots,n\}$, $C$ can be chosen proportional (i.e. up to a universal constant) to the maximum of the sub-Gaussian norms of $\{A_{1j}y_1 - \mathbf{E}[A_{1j}y_1]\}_{j=1}^n$ [3]. In most cases, however, it is involved to compute the sub-Gaussian norm exactly. However, it is well-known that for a zero-mean Gaussian random variable, the sub-Gaussian norm is proportional to its standard deviation; the precise value of the proportionality constant is not relevant to our analysis. In the sequel, we thus resort to a normal approximation as $|x_j^*| \to 0$, $j \in [n]$, $m \to \infty$, and evaluate the standard deviation of the limiting distribution. For this purpose, we derive the pdf $f_j$ of the random variables $A_{1j}y_1$, $j \in [n]$. Setting $X_j = A_{1j}$, $Y = y_1$, and using a well-known expression for the pdf of a product of random variables (cf. [2], §4.7), we obtain that

$$f_j(z) = \sum_{q\in\mathrm{range}(Q)} \frac{1}{|q|} f_{X_j,Y}(z/q, q)$$

$$= \sum_{q\in\mathrm{range}(Q)} \frac{1}{|q|} f_{X_j}(z/q) \, \mathbf{P}(Y = q | X_j = z/q)$$

$$= \sum_{k=1}^{K} \frac{1}{\mu_k} \phi(z/\mu_k) \left\{ \mathbf{P}\left(Y = \mu_k | X_j = z/\mu_k\right) + \mathbf{P}\left(Y = -\mu_k | X_j = z/-\mu_k\right) \right\}.$$

In the second line, the joint density $f_{X_j,Y}$ of $(X_j, Y)$ is factorized into the marginal density of $X_j$ and the conditional density (here discrete) of $Y$ given $X_j$. In the third line, we use that the range of $Q$ is $\{-\mu_K, \ldots, -\mu_1, \mu_1, \ldots, \mu_K\}$ and that $\phi(x) = \phi(-x)$ for all $x \in \mathbb{R}$. We now derive expressions for the conditional probabilities inside the curly brackets. Recall that $Y = \mu_k$ if and only if $\overline{Y} := \langle a_1, x^* \rangle + \sigma \varepsilon_1 \in (t_{k-1}, t_k)$, $k \in [K]$. We need to compute the probabilities of the events $\{\overline{Y} \in (t_{k-1}, t_k) | X_j = z/\mu_k\}$ and $\{\overline{Y} \in (-t_k, -t_{k-1}) | X_j = z/-\mu_k\}$. Note that $(X_j, \overline{Y})$ follow a bivariate Gaussian distribution with mean zero and the following second moments: $\mathrm{Var}(X_j) = 1$, $\mathrm{Var}(\overline{Y}) = 1 + \sigma^2$, $\mathrm{Cov}(X_j, \overline{Y}) = x_j^*$. Denoting $\widetilde{\rho} = x_j^*/\sqrt{1+\sigma^2}$ and making use of closed form expressions for the two conditional distributions (see e.g. [1]) associated with a bivariate Gaussian distribution, we obtain for any $k \in [K]$

$$\mathbf{P}\left(Y = \mu_k | X_j = z/\mu_k\right) = \mathbf{P}(\overline{Y} \in (t_{k-1}, t_k)|X_j = z/\mu_k)$$
$$= \Phi\left(\frac{t_k - \widetilde{\rho}\sqrt{1+\sigma^2}(z/\mu_k)}{\sqrt{1-\widetilde{\rho}^2}\sqrt{1+\sigma^2}}\right) - \Phi\left(\frac{t_{k-1} - \widetilde{\rho}\sqrt{1+\sigma^2}(z/\mu_k)}{\sqrt{1-\widetilde{\rho}^2}\sqrt{1+\sigma^2}}\right),$$

Likewise, for any $k \in [K]$ we have

$$\mathbf{P}\left(Y = -\mu_k | X_j = z/-\mu_k\right) = \mathbf{P}(\overline{Y} \in (-t_k, -t_{k-1})|X_j = z/-\mu_k)$$
$$= \Phi\left(\frac{-t_{k-1} - \widetilde{\rho}\sqrt{1+\sigma^2}(-z/\mu_k)}{\sqrt{1-\widetilde{\rho}^2}\sqrt{1+\sigma^2}}\right) - \Phi\left(\frac{-t_k - \widetilde{\rho}\sqrt{1+\sigma^2}(-z/\mu_k)}{\sqrt{1-\widetilde{\rho}^2}\sqrt{1+\sigma^2}}\right)$$
$$= \Phi\left(\frac{t_k - \widetilde{\rho}\sqrt{1+\sigma^2}(z/\mu_k)}{\sqrt{1-\widetilde{\rho}^2}\sqrt{1+\sigma^2}}\right) - \Phi\left(\frac{t_{k-1} - \widetilde{\rho}\sqrt{1+\sigma^2}(z/\mu_k)}{\sqrt{1-\widetilde{\rho}^2}\sqrt{1+\sigma^2}}\right),$$

using that $\Phi(-x) = 1 - \Phi(x)$ for all $x \in \mathbb{R}$. Altogether, we conclude that for all $j \in [n]$

$$f_j(z) = \sum_{k=1}^K \frac{1}{\mu_k} \phi(z/\mu_k)\, 2\left\{\Phi\left(\frac{t_k - \widetilde{\rho}\sqrt{1+\sigma^2}(z/\mu_k)}{\sqrt{1-\widetilde{\rho}^2}\sqrt{1+\sigma^2}}\right) - \Phi\left(\frac{t_{k-1} - \widetilde{\rho}\sqrt{1+\sigma^2}(z/\mu_k)}{\sqrt{1-\widetilde{\rho}^2}\sqrt{1+\sigma^2}}\right)\right\}.$$

Now note that as $|x_j^*| \to 0$, all $f_j$s converge pointwise to

$$f_0(x) = \sum_{k=1}^K \frac{1}{\mu_k} \phi(x/\mu_k)\left\{2(\Phi(t_k/\sqrt{1+\sigma^2}) - \Phi(t_{k-1}/\sqrt{1+\sigma^2}))\right\}$$
$$= \sum_{k=1}^K \mathbf{P}(|\widetilde{g}| \in \mathcal{R}_k(\mathbf{t}))\frac{1}{\mu_k}\phi(x/\mu_k), \quad \widetilde{g} \sim N(0, 1+\sigma^2)$$
$$= \sum_{k=1}^K \alpha_k(\mathbf{t})\frac{1}{\mu_k}\phi(x/\mu_k).$$

with $\boldsymbol{\alpha}(\mathbf{t})$ as defined in Lemma 2. Observe that $f_0$ equals the density of a Gaussian scale mixture with mixture proportions $\boldsymbol{\alpha}(\mathbf{t})$ and scales $\{\mu_k\}_{k=1}^K$. The standard deviation of this distribution is given by $\sqrt{\langle \boldsymbol{\alpha}(\mathbf{t}), \boldsymbol{\mu} \odot \boldsymbol{\mu} \rangle}$.

In light of the above, we conclude that as $|x_j^*| \to 0$, $A_{1j}y_1 - \mathbf{E}[A_{1j}y_1]$ converges to the Gaussian scale mixture with density $f_0$, $j \in [n]$. By the central limit theorem, $\sqrt{m}(\eta_j - \mathbf{E}[\eta_j])$ converges to a Gaussian distribution with standard deviation $\sqrt{\langle \boldsymbol{\alpha}(\mathbf{t}), \boldsymbol{\mu} \odot \boldsymbol{\mu} \rangle}$ as $m \to \infty$, $j \in [n]$. Consequently, the sub-Gaussian norm of $\sqrt{m}(\eta_j - \mathbf{E}[\eta_j])$ is proportional to $\sqrt{\langle \boldsymbol{\alpha}(\mathbf{t}), \boldsymbol{\mu} \odot \boldsymbol{\mu} \rangle}$ as $m \to \infty$, $j \in [n]$. $\qquad\square$

# G  Proof of Theorem 1

*Proof.* Consider the optimization problem

$$\min_{\mathbf{t}, \boldsymbol{\mu}} \Omega_b(\mathbf{t}, \boldsymbol{\mu}) = \min_{\mathbf{t}, \boldsymbol{\mu}} \frac{\Psi_b(\mathbf{t}, \boldsymbol{\mu})}{\lambda_b(\mathbf{t}, \boldsymbol{\mu})}.$$

By Lemma 2 and Lemma 3, the above minimization problem is equivalent to

$$\min_{\mathbf{t}, \boldsymbol{\mu}} R(\mathbf{t}, \boldsymbol{\mu}), \quad R(\mathbf{t}, \boldsymbol{\mu}) = \frac{\sqrt{\langle \boldsymbol{\alpha}(\mathbf{t}), \boldsymbol{\mu} \odot \boldsymbol{\mu} \rangle}}{\langle \boldsymbol{\alpha}(\mathbf{t}), \boldsymbol{E}(\mathbf{t}) \odot \boldsymbol{\mu} \rangle}, \tag{3}$$

where the term $\sigma^2 + 1$ in $\lambda_b$ has been dropped as it does not depend on $\mathbf{t}$ or $\boldsymbol{\mu}$. We start by claiming that

$$R(\mathbf{t}, \boldsymbol{\mu}) \geq \frac{1}{\sqrt{\langle \boldsymbol{\alpha}(\mathbf{t}), \boldsymbol{E}(\mathbf{t}) \odot \boldsymbol{E}(\mathbf{t}) \rangle}} \tag{4}$$

for all $\boldsymbol{\mu}$ with distinct, non-zero entries. The above lower bound is attained by choosing $\boldsymbol{\mu} \propto \boldsymbol{E}(\mathbf{t})$ (note that the minimizing $\boldsymbol{\mu}$ is only defined up to a positive constant as $R(\mathbf{t}, c\boldsymbol{\mu}) = R(\mathbf{t}, \boldsymbol{\mu})$ for all $c > 0$). Inequality (4) follows from the Cauchy-Schwarz inequality. Denote by $\boldsymbol{A}(\mathbf{t})$ the diagonal matrix whose diagonal is given by the entries of $\boldsymbol{\alpha}(\mathbf{t})$. We then have

$$\langle \boldsymbol{\alpha}(\mathbf{t}), \boldsymbol{E}(\mathbf{t}) \odot \boldsymbol{\mu} \rangle = \left\langle \boldsymbol{A}^{1/2}(\mathbf{t}) \boldsymbol{E}(\mathbf{t}), \boldsymbol{A}^{1/2}(\mathbf{t}) \boldsymbol{\mu} \right\rangle$$

$$\leq \sqrt{\left\langle \boldsymbol{A}^{1/2}(\mathbf{t}) \boldsymbol{E}(\mathbf{t}), \boldsymbol{A}^{1/2}(\mathbf{t}) \boldsymbol{E}(\mathbf{t}) \right\rangle} \sqrt{\left\langle \boldsymbol{A}^{1/2}(\mathbf{t}) \boldsymbol{\mu}, \boldsymbol{A}^{1/2}(\mathbf{t}) \boldsymbol{\mu} \right\rangle}$$

with equality holding if and only if

$$\boldsymbol{A}^{1/2}(\mathbf{t}) \boldsymbol{E}(\mathbf{t}) = c \boldsymbol{A}^{1/2}(\mathbf{t}) \boldsymbol{\mu} \Leftrightarrow \boldsymbol{E}(\mathbf{t}) = c \boldsymbol{\mu},$$

for some $c > 0$, where the above $\Leftrightarrow$ follows from the fact that the entries of $\mathbf{t}$ are required to be distinct so that the matrix $\boldsymbol{A}^{1/2}$ is regular. We conclude that

$$\min_{\mathbf{t}, \boldsymbol{\mu}} R(\mathbf{t}, \boldsymbol{\mu}) = \min_{\mathbf{t}} R(\mathbf{t}, \boldsymbol{E}(\mathbf{t})) = \min_{\mathbf{t}} \frac{1}{\sqrt{\langle \boldsymbol{\alpha}(\mathbf{t}), \boldsymbol{E}(\mathbf{t}) \odot \boldsymbol{E}(\mathbf{t}) \rangle}}. \tag{5}$$

We will now show that the above minimization problem in $\mathbf{t}$ is equivalent to the $b$-bit Lloyd-Max quantization problem of a random variable $h \sim N(0, 1 + \sigma^2)$, which we re-state here for convenience:

$$\min_{\mathbf{t}, \boldsymbol{\mu}} \mathbf{E}[\{h - Q(h; \mathbf{t}, \boldsymbol{\mu})\}^2] = \min_{\mathbf{t}, \boldsymbol{\mu}} \mathbf{E}[\{h - \mathrm{sign}(h) \sum_{k=1}^{K} \mu_k I(|h| \in \mathcal{R}_k(\mathbf{t}))\}^2] \tag{6}$$

For the above problem, it is not hard to see that for any fixed choice of $\mathbf{t}$, the minimizing $\boldsymbol{\mu}^*(\mathbf{t})$ is given by $\mu_k^*(\mathbf{t}) = \mathbf{E}[h | h \in \mathcal{R}_k(\mathbf{t})] = E_k(\mathbf{t})$, $k \in [K]$, where we recall that $E_k(\mathbf{t})$ is the $k$-th component of $\boldsymbol{E}(\mathbf{t})$ as appearing above. To finish the proof of the first part of the Theorem 1, it thus remains to show that after substituting $\boldsymbol{\mu}^*(\mathbf{t})$ back into (6), the resulting minimization problem in $\mathbf{t}$ is equivalent to (5). We have

$$\min_{\mathbf{t}} \mathbf{E} \left[ \left\{ h - \mathrm{sign}(h) \sum_{k=1}^{K} I(|h| \in \mathcal{R}_k(\mathbf{t})) \mathbf{E}[h | h \in \mathcal{R}_k(\mathbf{t})] \right\}^2 \right]$$

$$= 2 \min_{\mathbf{t}} \mathbf{E} \left[ \sum_{k=1}^{K} I(h \in \mathcal{R}_k(\mathbf{t}))(h - \mathbf{E}[h | h \in \mathcal{R}_k(\mathbf{t})])^2 \right]$$

$$= 2 \min_{\mathbf{t}} \mathbf{E} \left[ \sum_{k=1}^{K} I(h \in \mathcal{R}_k(\mathbf{t})) \left\{ h^2 - 2h \, \mathbf{E}[h | h \in \mathcal{R}_k(\mathbf{t})] + \mathbf{E}[h | h \in \mathcal{R}_k(\mathbf{t})]^2 \right\} \right]$$

$$= \mathbf{E}[h^2] + 2 \min_{\mathbf{t}} \left\{ -2 \sum_{k=1}^{K} \mathbf{E}[h | h \in \mathcal{R}_k(\mathbf{t})] \, \mathbf{E}[I(h \in \mathcal{R}_k(\mathbf{t}))h] + \right.$$

$$\left. + \sum_{k=1}^{K} \mathbf{P}(h \in \mathcal{R}_k(\mathbf{t})) \, \mathbf{E}[h | h \in \mathcal{R}_k(\mathbf{t})]^2 \right\}$$

$$= 1 + \min_{\mathbf{t}} - \sum_{k=1}^{K} \mathbf{E}[h | h \in \mathcal{R}_k(\mathbf{t})]^2 \, \mathbf{P}(|h| \in \mathcal{R}_k(\mathbf{t}))$$

$$= 1 + \min_{\mathbf{t}} - \langle \boldsymbol{E}(\mathbf{t}) \odot \boldsymbol{E}(\mathbf{t}), \boldsymbol{\alpha}(\mathbf{t}) \rangle,$$

which establishes the equivalence to (5) as claimed.

We now prove the second part of the Theorem. Denote by $\mathbf{t}_0^*$ the Lloyd-Max optimal thresholds for $\sigma = 0$, i.e. for a $N(0,1)$ variable. Clearly, $\mathbf{t}^* = \mathbf{t}_\sigma^* = \sqrt{1 + \sigma^2}\mathbf{t}_0^*$ for any $\sigma > 0$. Evaluating $\Omega_b(\mathbf{t}^*, \boldsymbol{\mu}^*)$, we obtain in view of (5)

$$\Omega_b(\mathbf{t}^*, \boldsymbol{\mu}^*) \propto \frac{1 + \sigma^2}{\sqrt{\langle \boldsymbol{\alpha}(\mathbf{t}^*), \boldsymbol{E}(\mathbf{t}^*) \odot \boldsymbol{E}(\mathbf{t}^*) \rangle}}$$

$$= \frac{1 + \sigma^2}{\sqrt{\langle \boldsymbol{\alpha}(\mathbf{t}_0^*\sqrt{1 + \sigma^2}), \boldsymbol{E}(\mathbf{t}_0^*\sqrt{1 + \sigma^2}) \odot \boldsymbol{E}(\mathbf{t}_0^*\sqrt{1 + \sigma^2}) \rangle}}$$

Evaluating the expression in the denominator, we obtain that

$$\alpha_k(\mathbf{t}_0^*\sqrt{1 + \sigma^2}) = \mathbf{P}(|\widetilde{g}| \in \mathcal{R}_k(\mathbf{t}_0^*\sqrt{1 + \sigma^2})) = \mathbf{P}(|g| \in \mathcal{R}_k(\mathbf{t}_0^*)), \ k \in [K],$$

where $\widetilde{g} \sim N(0, 1 + \sigma^2)$, $g \sim N(0,1)$. Moreover, with the help of Lemma E.3

$$\boldsymbol{E}(\mathbf{t}_0^*\sqrt{1 + \sigma^2}) = \left( \mathbf{E}[\widetilde{g}|\widetilde{g} \in \mathcal{R}_k(\mathbf{t}_0^*\sqrt{1 + \sigma^2})] \right)_{k=1}^K$$

$$= \sqrt{1 + \sigma^2}\left( \mathbf{E}[g|g \in \mathcal{R}_k(\mathbf{t}_0^*)] \right)_{k=1}^K.$$

Putting together the pieces, we obtain that

$$\Omega_b(\mathbf{t}^*, \boldsymbol{\mu}^*) \propto \frac{1 + \sigma^2}{\sqrt{\langle \boldsymbol{\alpha}_0(\mathbf{t}_0^*), (1 + \sigma^2)\boldsymbol{E}_0(\mathbf{t}_0^*) \odot \boldsymbol{E}_0(\mathbf{t}_0^*) \rangle}} = \frac{\sqrt{1 + \sigma^2}}{\sqrt{\lambda_{b,0}(\mathbf{t}_0^*, \boldsymbol{\mu}_0^*)}}$$

where the $\boldsymbol{\alpha}_0(\mathbf{t})$, $\boldsymbol{E}_0(\mathbf{t})$ and $\lambda_{b,0}(\mathbf{t}, \boldsymbol{\mu})$ refer to the definitions of $\boldsymbol{\alpha}(\mathbf{t})$, $\boldsymbol{E}(\mathbf{t})$, $\lambda_b(\mathbf{t}, \boldsymbol{\mu})$ for $\sigma = 0$. This completes the proof. $\qquad\square$

# H   Derivations for the paragraph 'Beyond additive noise'

We fix $\sigma = 0$ and the corresponding Lloyd-Max optimal choices $\mathbf{t} = \mathbf{t}_0^*$, $\boldsymbol{\mu} = \boldsymbol{\mu}_0^*$ so that $\mu_k = \mathbf{E}[g|g \in \mathcal{R}_k]$, $g \sim N(0,1)$ with $\mathcal{R}_k = \mathcal{R}_k(\mathbf{t}_0^*)$, $k \in [K]$.

*Mechanism (I)*

In order to evaluate $\lambda = \lambda_{b,p}$, we first need to derive an expression for the corresponding map $\theta$. Recalling Definition E.1, we have

$$\mathbf{E}[y_1|a_1] = (1 - p)\sum_{k=1}^{2^b}\widetilde{\mu}_k I(\langle a_1, x^* \rangle \in \mathcal{Q}_k) + p\frac{1}{2^b - 1}\sum_{k=1}^{2^b}\widetilde{\mu}_k I(\langle a_1, x^* \rangle \notin \mathcal{Q}_k)$$

and thus

$$\theta(z) = (1 - p)\sum_{k=1}^{2^b}\widetilde{\mu}_k I(z \in \mathcal{Q}_k) + p\frac{1}{2^b - 1}\sum_{k=1}^{2^b}\widetilde{\mu}_k I(z \notin \mathcal{Q}_k)$$

It follows that for $g \sim N(0,1)$

$$
\begin{aligned}
\lambda_{b,p} = \mathbf{E}[g\,\theta(g)] &= \sum_{k=1}^{2^b} \widetilde{\mu}_k \left\{ (1-p)\,\mathbf{E}\left[gI(g \in \mathcal{Q}_k)\right] + p\frac{1}{2^b-1}\,\mathbf{E}[gI(g \notin \mathcal{Q}_k)] \right\} \\
&= \sum_{k=1}^{2^b} \widetilde{\mu}_k \left\{ (1-p)\,\mathbf{E}\left[gI(g \in \mathcal{Q}_k)\right] + p\frac{1}{2^b-1}\,\mathbf{E}[g(1 - I(g \in \mathcal{Q}_k))] \right\} \\
&= \sum_{k=1}^{2^b} \widetilde{\mu}_k \left\{ (1-p) - \frac{p}{2^b-1} \right\} \mathbf{E}\left[gI(g \in \mathcal{Q}_k)\right] \\
&= \sum_{k=1}^{K} \mathbf{P}(|g| \in \mathcal{R}_k)\,\mathbf{E}[g|g \in \mathcal{R}_k]^2 \left\{ (1-p) - \frac{p}{2^b-1} \right\} \\
&= \langle \boldsymbol{\alpha}_0(\mathbf{t}_0^*), \boldsymbol{E}_0(\mathbf{t}_0^*) \odot \boldsymbol{E}_0(\mathbf{t}_0^*) \rangle \left\{ (1-p) - \frac{p}{2^b-1} \right\} \\
&= \lambda_{b,0} \left\{ (1-p) - \frac{p}{2^b-1} \right\},
\end{aligned}
$$

where $\boldsymbol{\alpha}_0(\mathbf{t}_0^*)$ and $\boldsymbol{E}_0(\mathbf{t}_0^*)$ are defined at the end of the preceding proof. From the last expression we deduce the breakdown point $\bar{p}_b = 1 - 1/2^b$.
For evaluating $\Psi_{b,p}$ (up to a positive constant), we make use of the asymptotic expression $\Psi_{b,0} \propto \sqrt{\langle \boldsymbol{\alpha}(\mathbf{t}), \boldsymbol{\mu} \odot \boldsymbol{\mu} \rangle}$ derived in Lemma 3. The only thing that changes under Mechanism (I) are the probabilities $\boldsymbol{\alpha}(\mathbf{t})$ which become

$$
\alpha_k(\mathbf{t}) = \mathbf{P}(|g| \in \mathcal{R}_k(\mathbf{t}))(1-p) + \frac{p}{2^{b-1}}\sum_{l \neq k} \mathbf{P}(|g| \in \mathcal{R}_l(\mathbf{t})), \quad k \in [K].
$$

*Mechanism (II)*
Following the same route as for Mechanism (I), one derives

$$
\theta(z) = (1-p)\sum_{k=1}^{2^b} \widetilde{\mu}_k I(z \in \mathcal{Q}_k) + p\left\{ -\mu_K I(z \geq 0) + \mu_K I(z < 0) \right\}
$$

and accordingly for $g \sim N(0,1)$

$$
\begin{aligned}
\lambda_{b,p} = \mathbf{E}[g\,\theta(g)] &= (1-p)\sum_{k=1}^{K} \mathbf{P}(|g| \in \mathcal{R}_k)\,\mathbf{E}[g|g \in \mathcal{R}_k]^2 - p\mu_K\,\mathbf{E}[g|g > 0] \\
&= (1-p)\lambda_{b,0} - p\mu_K\sqrt{2/\pi}
\end{aligned}
$$

so that the breakdown points results as $\bar{p}_b = \lambda_{b,0}/(\lambda_{b,0} + \mu_K\sqrt{2/\pi})$. As for Mechanism (II), $\Psi_{b,p}$ is obtained by evaluating the changes in $\boldsymbol{\alpha}(\mathbf{t})$. We have

$$
\alpha_k(\mathbf{t}) = (1-p)\,\mathbf{P}(|g| \in \mathcal{R}_k(\mathbf{t})), \quad k \in [K-1],
$$

$$
\alpha_K(\mathbf{t}) = p\sum_{k=1}^{K-1} \mathbf{P}(|g| \in \mathcal{R}_k(\mathbf{t})) + \mathbf{P}(|g| \in \mathcal{R}_K(\mathbf{t})).
$$

# I  Proof of Proposition 4

*Proof.* In the sequel, we derive tail bounds of the form

$$
\begin{aligned}
\mathbf{P}(\widehat{\psi} \geq (1+\varepsilon)\psi^*) &\leq \exp(-cm\varepsilon^2), \\
\mathbf{P}(\widehat{\psi} \leq (1-\varepsilon)\psi^*) &\leq \exp(-2cm\varepsilon^2).
\end{aligned}
$$

for $\varepsilon \in (0,1)$ and $c = 2\{\phi'(t/\psi^*)\}^2$. This implies that the probability of the event

$$\left| \frac{\widehat{\psi}}{\psi^*} - 1 \right| > \varepsilon$$

is upper bounded by $2\exp(-cm\varepsilon^2)$.

**1) Upper tail**

$$
\begin{aligned}
\mathbf{P}(\widehat{\psi} \geq (1+\varepsilon)\psi^*) &= \mathbf{P}\left( \frac{t_1}{\Phi^{-1}\left(\frac{1}{2}(1 + \frac{m_1}{m})\right)} \geq (1+\varepsilon)\psi^* \right) \\
&= \mathbf{P}\left( \frac{m_1}{m} \leq 2\Phi\left( \frac{t_1}{(1+\varepsilon)\psi^*} \right) - 1 \right) \\
&= \mathbf{P}\left( \frac{m_1}{m} - \mathbf{E}\left[\frac{m_1}{m}\right] \leq 2\left\{ \Phi\left( \frac{t_1}{(1+\varepsilon)\psi^*} \right) - \Phi\left( \frac{t_1}{\psi^*} \right) \right\} \right)
\end{aligned}
$$

We have

$$
\begin{aligned}
\Phi\left( \frac{t_1}{(1+\varepsilon)\psi^*} \right) - \Phi\left( \frac{t_1}{\psi^*} \right) &= -\int_{t_1/(\psi^*(1+\varepsilon))}^{t_1/\psi^*} \phi(u)\, du \\
&\leq -\phi(t_1/\psi^*) \frac{t_1}{\psi^*} \frac{\varepsilon}{\varepsilon + 1} \\
&\leq -\phi(t_1/\psi^*) \frac{t_1}{\psi^*} \frac{\varepsilon}{2} = \phi'(t_1/\psi^*)\frac{\varepsilon}{2}.
\end{aligned}
$$

for $\varepsilon \in (0,1)$. Thus

$$\mathbf{P}(\widehat{\psi} \geq (1+\varepsilon)\psi^*) \leq \mathbf{P}\left( \frac{m_1}{m} - \mathbf{E}\left[\frac{m_1}{m}\right] \leq \varepsilon\phi'(t_1/\psi^*) \right)$$

**2) Lower tail**

Similarly, we obtain that

$$\mathbf{P}(\widehat{\psi} \leq (1-\varepsilon)\psi^*) \leq \mathbf{P}\left( \frac{m_1}{m} - \mathbf{E}\left[\frac{m_1}{m}\right] \geq 2\left\{ \Phi\left( \frac{t_1}{(1-\varepsilon)\psi^*} \right) - \Phi\left( \frac{t_1}{\psi^*} \right) \right\} \right)$$

We have

$$\Phi\left( \frac{t_1}{(1-\varepsilon)\psi^*} \right) - \Phi\left( \frac{t_1}{\psi^*} \right) = \int_{t_1/\psi^*}^{t_1/(\psi^*(1-\varepsilon))} \phi(u)\, du \quad \geq \phi(t_1/\psi^*)\frac{t_1}{\psi^*}\frac{\varepsilon}{1-\varepsilon} \geq -\phi'(t_1/\psi^*)\varepsilon.$$

Thus,

$$\mathbf{P}(\widehat{\psi} \leq (1-\varepsilon)\psi^*) \leq \mathbf{P}\left( \frac{m_1}{m} - \mathbf{E}\left[\frac{m_1}{m}\right] \leq 2\varepsilon(-\phi'(t_1/\psi^*)) \right).$$

Note that $m_1$ is a Binomial random variable. Applying Hoeffding's inequality to **1)** and **2)**, we obtain that

$$
\begin{aligned}
\mathbf{P}(\widehat{\psi} \geq (1+\varepsilon)\psi^*) &\leq \exp\left(-2m\varepsilon^2\{\phi'(t_1/\psi^*)\}^2\right) \\
\mathbf{P}(\widehat{\psi} \leq (1-\varepsilon)\psi^*) &\leq \exp\left(-4m\varepsilon^2\{\phi'(t_1/\psi^*)\}^2\right).
\end{aligned}
$$

which proves the claim made above. $\qquad\square$

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

[Supplementary Material 2]

# Supplement to '$b$-bit Marginal Regression': full set of experimental results

# Empirical verification of the analysis of Section 3

$\sigma = 0$

$\sigma = 1$

$\sigma = 2$

# Estimation of the scale and the noise level

## Estimation of the scale parameter

**Estimation of the noise level**

# *b*-bit Marginal Regression and alternative recovery algorithms

$b = 1$, $\sigma = 0$

$b = 2,\ \sigma = 0$

$b = 1,\ \sigma = 1$

$b = 2,\ \sigma = 1$

$b = 1,\ \sigma = 2$

$b = 2,\ \sigma = 2$