[Reviews · NeurIPS 2015]

Submitted by Assigned_Reviewer_1

The paper presents a method for b-bit compressive sensing, i.e., estimate x^* from y and A, where y=Q(Ax^*), Q maps a real number into a b-bit string. The algorithm uses the largest entries of A^Ty to estimate x^*/||x^*||. ||x^*|| is estimated using an MLE estimate.

The main message of the paper is that two-bit compressive sensing is most efficient in terms of estimating x^* accurately.

The overall message seems to be interesting and the empirical results seem to verify it. The paper is well written as well and fun to read.

However, the method and the claims of this paper are not rigorous and needs some more work.

a) The method just uses one step process of estimating x^* (i.e. uses A^Ty). However, such a method is known to require significantly larger sample complexity and in practice as well the method is significantly worse than the other method like IHT or L1 minimization (authors observe the same in their experiments as well).

b) Due to the weak method, the error bounds for the method are also significantly worse than the existing results for compressive sensing (both standard CS and one-bit style CS). For example, the given method naturally cannot do exact recovery even when the measurements are noiseless. Moreover, the recovery is not universal that for each x^*, a new A has to be sampled. This again is not very realistic and interesting scenario to study.

c) The claim that b=2-bit CS has the best sample complexity is quite confusing. Because i) the method considered itself is suboptimal (as described above) ii) the analysis of the method gives only an upper bound on samples and authors claim that "inspection of proof indicates that the dependence is tight" is a bit vague and does not imply any lower bound on the number of samples required iii) the analysis of the upper bound as well as MLE estimate of ||x||_* is also not complete and requires computing certain quantities using experiments.

Overall the paper contains an interesting claim but the claim is not fully justified. Especially, the method considered to make the claim is quite weak and does not even work well in practice (in reasonably low-noise scenarios).
Summary: See below.

Submitted by Assigned_Reviewer_2

The authors extended the 1-bit CS to b-bit marginal regression problem. The approach is novel and extension numerical experiments were done to verify the derived theoretical properties. The paper seems to bridge the gap between the 1-bit CS and traditional high precision CS.

The writing of the paper is clear. However, it would be useful if the authors could discuss the computational cost of the proposed algorithm to other methods.
Summary: The authors extended the 1-bit CS to b-bit marginal regression problem. The approach looks novel and extension numerical experiments were done to verify the derived theoretical properties.

Submitted by Assigned_Reviewer_3

The paper considers a theoretically interesting problem of sparse signal recovery from m linear measurements quantized to b bits. The authors argue that what matters is total bits bm rather than number of measurements m. They propose a b-bit Marginal regression as the recovery algorithm. They theoretically derive the error bounds leading to the computable expressions explaining the m and b trade-off. They show some interesting observations like: b=1 is optimal for finding the support while b=2 is sufficient for finding support as well as norm. Synthetic experiments validate the theory.

I really liked the paper. The theory in the paper is elegant, novel and interesting. The work in some sense generalizes the 1 bit compressive sensing results. The bounds provided on the error seems tight as validated by the experiments. The observations like 1 bit is enough to recover the support and 2 is enough to recover the full signal along with the magnitude are very interesting and practically useful.

The paper is very dense and intensive to read mainly because of the theoretical nature. I would suggest authors to spend some space on explaining main results like Proposition 2 and Theorem 1 more clearly.

The theory in the paper seems original. The results explaining the m and b trade-off are novel and significant practically.
Summary: The theory presented in the paper for understanding the trade-off between quantization and measurements m is novel, elegant and practically useful. The error bounds leads to expressions explaining the trade-off and giving interesting observations like: b=1 is optimal for finding the support while b=2 is sufficient for finding support as well as norm.

Submitted by Assigned_Reviewer_4

The paper proposes to use marginal regression for sparse recovery from measurements quantized to b bits and studies the trade-off between the number of measurements and the number of bits per measurements. For marginal regression, the optimal choice seems to be to use 2 bits, which allows to estimate the signal's scale too. The authors establish some interesting results, but at times fail to sufficiently motivate their assumptions, cf Lemma 3. Furthermore, they focus on adversarial bin flips errors and not on random bin flip errors, which from the table in page 5 seem to be better addressed by using more bits. Finally, it is not clear how it is possible to obtain an estimate of x^*_u without knowing a priori how to set up the quantization bins. The authors should also explore b >= 3, to see whether more bits may be beneficial for the other algorithms compared against in section 5.
Summary: An interesting paper that addresses the grey zone between 1-bit and infinite precision sparse recovery. The paper could be improved by better motivating the assumptions and by extending the numerical experiments to b >= 3.

Author Feedback
Author rebuttal: Thanks for your comments and suggestions, and for considering our responses below. Your concerns have been rephrased for brevity.

REVIEWERS 1 and 5:
Tightness of the error bound (11) is not shown.

RESPONSE:
We can prove that the dependence of (11) on Omega_b = Psi_b/lambda_b is in fact tight. Our analysis is thus meaningful as it compares Omega_b/Omega_b' for
different b,b'. Our experiments confirm this.

REVIEWER 1:
b-bit Marginal Regression should be compared to approaches to b-bit CS that treat quantization as additive error and then apply a lasso-type formulation.

RESPONSE:
Thanks for pointing this out. The error bounds of these approaches depend on the quantization error in an additive way, whereas the dependence is multiplicative for our method (factor 1/lambda). For fixed b, the error of these
alternative approaches does not vanish as m goes to infinity, but vanishes as b goes infinity (for m not too small), unlike our method. We will include a discussion as well as an empirical comparison to at least one of these approaches in a revised version.

We also thank Reviewer 1 for spotting a wrong reference and a typo.

REVIEWER 2:
The authors might discuss the computational cost of the proposed algorithm relative to others.

RESPONSE:
The computational cost behaves rather favorably compared to others. This will be stated more explicitly.

REVIEWER 3:
I suggest that the authors spend more space on explaining the main results of the paper.

RESPONSE:
We will keep the introductory part in Sec.2 more brief to free up some space.

REVIEWER 4:
It is not clear how to set up the quantization bins if the norm of the signal is not known a priori.

RESPONSE:
That's indeed an issue in practice. To deal with it, one might use the scale estimator in Sec.4 of the paper adaptively in a way such that the estimate of the scale is updated after each measurement.

REVIEWER 4:
b >= 3 should be explored in the experiments.

RESPONSE:
For space reasons, we have confined ourselves to the transition from b=1 to b=2. For b=3, the results essentially mirror the contrast between a noiseless and a noisy setting already shown in the paper. We also would like to refer to [9,12].

REVIEWER 5:
The recovery algorithm considered in the paper is suboptimal. The error bounds and the performance in practice is worse than those of other methods.

RESPONSE:
The reviewer is right inasmuch as high precision (large b) and low-noise scenarios are concerned. E.g., as also pointed out by the reviewer, Marginal Regression does not achieve exact recovery for infinite precision and zero noise, and - as shown in the paper - the empirical performance is inferior to other algorithms for b=1,2 and low noise.

On the other hand, the recovery algorithm considered in our paper, which builds on a proposal of Plan/Vershynin, has received a lot of interest in the literature in particular because

a) it is computationally convenient
b) it has reasonable theoretical guarantees

For example, IHT-type algorithms for b-bit measurements [9] satisfy a), but little seems to be known for b); the method in [10] for _noiseless_ 1-bit CS which uses an ell0-constraint has an error bound of O(s log(n)/m) which is better than O(\sqrt{s log(n)/m}) in our bound in Eq.(11), but that method is computationally not tractable.

The rate in (11) cannot be improved much in the case of additive Gaussian noise as the lower bound is of the order \sqrt{s log(n/s)/m} for full precision measurements.
As shown by our experiments, b-bit Marginal Regression compares favorably to several other methods in low-precision, moderate to high noise regimes. The last 2 points indicate to us that the method is a state-of-the-art competitor - at least in certain regimes of interest.

Moreover, the simplicity of the method is not only computationally beneficial, it is also key to our analysis of the trade-off between m and b with rather explicit results.